# Emergence and diversification of a highly invasive chestnut pathogen lineage across southeastern Europe

**Lea Stauber[1,2], Thomas Badet[2], Alice Feurtey[2,3], Simone Prospero[1], Daniel Croll[2]***

[1]Swiss Federal Institute for Forest, Snow and Landscape Research (WSL), Birmensdorf, Switzerland; [2]Laboratory of Evolutionary Genetics, Institute of Biology, University of Neuchâtel, Neuchâtel, Switzerland; [3]Plant Pathology, Institute of Integrative Biology, ETH Zürich, Zürich, Switzerland

**Abstract** Invasive microbial species constitute a major threat to biodiversity, agricultural production and human health. Invasions are often dominated by one or a small number of genotypes, yet the underlying factors driving invasions are poorly understood. The chestnut blight fungus *Cryphonectria parasitica* first decimated the North American chestnut, and a more recent outbreak threatens European chestnut stands. To unravel the chestnut blight invasion of southeastern Europe, we sequenced 230 genomes of predominantly European strains. Genotypes outside of the invasion zone showed high levels of diversity with evidence for frequent and ongoing recombination. The invasive lineage emerged from the highly diverse European genotype pool rather than a secondary introduction from Asia or North America. The expansion across southeastern Europe was mostly clonal and is dominated by a single mating type, suggesting a fitness advantage of asexual reproduction. Our findings show how an intermediary, highly diverse bridgehead population gave rise to an invasive, largely clonally expanding pathogen.

*For correspondence:
daniel.croll@unine.ch

**Competing interests:** The authors declare that no competing interests exist.

## Introduction

Over the past century, a multitude of invasive species have emerged as threats to forest and agricultural ecosystems worldwide (*Santini et al., 2013*; *Wingfield et al., 2010*). Increased human activities, such as global trade, enabled invasive species to cause economic damage through reduced agricultural production, degradation of ecosystems, and negative impacts on human health (*Marbuah et al., 2014*). A key group of invasive species in forests are fungal pathogens, which are often accidentally introduced via living plants and plant products (*Rossman, 2001*). To successfully colonize a new environment, fungal pathogens have to overcome several invasion barriers including effective dispersal abilities, changes in available hosts, competition with other fungi, and niche availability (*Gladieux et al., 2015*; *Hayes and Barry, 2008*). This may be achieved with plastic phenotypic changes followed by rapid genetic adaptation (*Garbelotto et al., 2015*). The invasive potential is also influenced by pre-adaptation arising outside of the invasion zone. Pre-adaptation can be a critical factor for environmental filtering by abiotic or host-related factors in the new environment (*Krasnov et al., 2015*). As the number of initial founders is often low, invasive populations are frequently of low genetic diversity, which reduces adaptive genetic variation (*Allendorf and Lundquist, 2003*; *Yang et al., 2012*). Yet, many fungal plant pathogen invasions were successful despite low genetic diversity within founding populations (*Fontaine et al., 2013*; *Raboin et al., 2007*; *Wuest et al., 2017*).

A major model explaining the successful expansion of invasive populations despite low initial genetic diversity is the so-called bridgehead effect (*Lombaert et al., 2010*). In this model, highly adapted lineages emerge through recombination among genotypes established in an area of first

introduction. Hence, the primary introduction serves as the bridgehead for a secondary and more expansive invasion. Although the bridgehead effect has been proposed as a scenario for many biological invasions (*Gau et al., 2013*; *van Boheemen et al., 2017*), empirical evidence for the creation of highly adaptive genotypes within bridgehead populations is still largely missing (*Bertelsmeier and Keller, 2018*). An alternative explanation suggests that primary bridgehead populations simply serve as repeated sources of inoculum for secondary invasions without selecting for the invasive (i.e., adapted) genotypes in situ (*Bertelsmeier and Keller, 2018*). This alternative scenario implies that initial populations are already composed of genotypes adapted to the new environment or have high phenotypic plasticity (*Bock et al., 2015*; *Gladieux et al., 2015*; *Vuković et al., 2019*). A conceptually similar model to the bridgehead effect was proposed for bacteria, where the success of epidemic clones would be contingent on the generation of recombinants in the original populations (*Smith et al., 1993*). Dissecting whether invasive species were preadapted to the new environment or gained adaptation through a bridgehead effect is crucial for effective containment strategies, but requires a deep sampling of genotypes during the early invasion process.

The pace of adaptive evolution leading to successful invasions is determined by several life history traits including the mating system (i.e., sexual, asexual, or mixed). Sexual reproduction can potentially promote the emergence of more virulent strains through the reshuffling of genetic variants, enhancing invasive success (*Philibert et al., 2011*). In contrast, a switch to asexual reproduction can limit genetic diversification and adaptive potential in newly introduced species (*Drenth et al., 2019*; *Taylor et al., 2015*). However, the low availability of mating partners at the invasion front can be a major cost for sexual reproduction. To avoid the cost of mate search, invasive fungal pathogens often switch from sexual to predominantly asexual reproduction (*Heitman et al., 2013*; *Suehs et al., 2004*). Consequently, asexual reproduction can promote rapid colonization and dissemination of adapted genotypes (*Gladieux et al., 2011*). Beyond invasive species, many fungal pathogens undergo asexual reproduction during the colonization of habitats, but they switch to sexual reproduction for the production of survival propagules prior to unfavorable environmental conditions (*Schoustra et al., 2010*). Populations of some highly successful invaders, such as the ascomycete *Ophiostoma novo-ulmi* causing Dutch elm disease (*Paoletti et al., 2006*), or the oomycete *Phytophthora ramorum* (*Grünwald et al., 2012*), are dominated by a single mating type. Low diversity in invasive lineages may be breached by secondary invasions introducing the opposite mating type as observed in *Phytophthora infestans* in Europe. Prior to the 1980s, only the presence of the A1 mating type was clearly established, followed by the secondary introduction of the A2 mating type leading to sexual reproduction (*Mariette et al., 2016*). Introgression from closely related species can also reintroduce a missing mating type. *O. novo-ulmi* acquired the missing mating type from *O. ulmi* (*Brasier and Webber, 2019*; *Paoletti et al., 2006*). Although switches in reproductive modes can be a key factor for invasion success (*Philibert et al., 2011*), mechanisms underlying such switches remain poorly understood (*Billiard et al., 2012*). Beyond the impact on genetic diversity, reproductive modes are often tied to specific environmental adaptations including the mode of spore dispersal. Sexual spores are often highly durable and can be wind dispersed over long distances (*Philibert et al., 2011*; *Rigling and Prospero, 2018*). Even rare long-distance dispersal events can contribute to the geographic expansion of invasive pathogens (*Prospero and Cleary, 2017*). In contrast, asexual spores serve mostly short distance dispersal in fungal pathogens with mixed reproduction (*Mehrotra, 2013*; *Prospero and Cleary, 2017*).

Some invasive pathogens were discovered to harbor specific characteristics in the architecture of the genome, which may have facilitated the generation of adaptive genetic variation. For instance, genomes of *Ophiostoma* species causing Dutch elm disease show signals of frequent hybridization and introgression events, giving rise to lineages with increased virulence or growth at higher temperatures (*Hessenauer et al., 2020*). Genomes of a number of crop pathogens including the oomycete *P. infestans* and the fungal pathogens *Leptosphaeria maculans* and *Zymoseptoria tritici* show compartmentalization into gene-dense and gene-poor regions with virulence factor being preferentially encoded in gene-poor regions (*Dong et al., 2015*; *Frantzeskakis et al., 2019*; *Torres et al., 2020*). The gene-poor regions are enriched in transposable elements (TEs), and the encoded virulence factors (i.e., effectors) show epigenetic regulation, rapid sequence turnover, and weak phylogenetic conservation (*Fouché et al., 2018*). In pathogen systems such as *Cryptococcus gattii* or *P. infestans*, copy number variation (CNV) has created highly successful outbreak lineages

(*Knaus et al., 2020*; *Steenwyk et al., 2016*). Beyond potential roles in adaptation, the genome structure of plant pathogens is likely influenced by neutral processes including migratory bottlenecks possibly affecting the activity of TEs (*Oggenfuss et al., 2020*). Whether virulence factors are associated with genome compartmentalization and how chromosomal rearrangements could underpin the creation of invasive pathogen genotypes remain largely unknown.

The ascomycete *Cryphonectria parasitica* (Murr.) Barr. is the causal agent of chestnut blight, a lethal bark disease of *Castanea* species (*Rigling and Prospero, 2018*). The pathogen is native to eastern Asia and was first observed in 1904 in North America on the American chestnut (*Castanea dentata*). In the following years, the disease rapidly spread throughout the distribution range of *C. dentata*, causing the vast decimation of this native tree species (*Elliott and Swank, 2008*). In 1938, the fungus was first detected on the European chestnut (*Castanea sativa*) near Genoa (Italy) and is now established in all major chestnut-growing countries in Europe (*Rigling and Prospero, 2018*). The damage to the European chestnut may have been reduced by the presence of the *Cryphonectria hypovirus 1* (CHV1), which acts as a biological control agent of chestnut blight (*Rigling and Prospero, 2018*). The virus can be transmitted both vertically to asexual spores (conidia) or horizontally through hyphal anastomoses between virus-infected and virus-free strains (*Heiniger and Rigling, 1994*). Hyphal anastomoses are controlled by a vegetative compatibility system, and the virus spreads most effectively between strains of the same vegetative compatibility type (*Cortesi et al., 2001*).

Population genetic analyses showed that the invasion of Europe occurred through multiple introductions both from native populations in Asia and from populations in North America (*Dutech et al., 2012*). Invasive European *C. parasitica* populations are characterized by lower vegetative compatibility diversity than North American and Asian populations (*Milgroom and Cortesi, 1999*). Furthermore, European populations exhibit lower recombination rates (*Dutech et al., 2010*; *González-Varela et al., 2011*; *Prospero and Rigling, 2012*). Mating in *C. parasitica* is controlled by a single mating type (MAT) locus (*Marra and Milgroom, 2001*), and natural populations can reproduce both sexually and asexually (*Marra et al., 2004*). Previous analyses of southeastern European *C. parasitica* populations based on sequence characterized amplified region (SCAR) markers suggested that the region was largely colonized by a single and likely asexual lineage also identified as S12 (*Milgroom et al., 2008*). The lineage belongs to the vegetative compatibility type EU-12. Within the distribution range of the lineage, sexual structures (i.e., perithecia) are rarely found (*Milgroom et al., 2008*; *Sotirovski et al., 2004*). Based on SCAR marker and field records, *Milgroom et al., 2008* suggested that the invasive S12 lineage originated in northern Italy and subsequently spread across southeastern Europe (*Avolio, 1978*; *Biraghi, 1946*; *Buccianti and Feliciani, 1966*; *Karadžić et al., 2019*; *Myteberi et al., 2013*; *Robin and Heiniger, 2001*). However, due to the low molecular marker resolution and challenges in relying on observational data, the origin, invasion route, and genetic diversification of *C. parasitica* in southeastern Europe remain largely unknown.

In this study, we sequenced complete genomes of a comprehensive collection of European *C. parasitica* strains with a fine-scaled sampling throughout the S12 invasion zone. Based on high-confidence genome-wide single nucleotide polymorphisms (SNPs), we identified the most likely origin and recapitulated the invasion process across southeastern Europe. We show that the invasive *C. parasitica* lineage arose through an intermediary, highly diverse bridgehead population. During the expansion, the lineage became dominated by a single mating type but retained the ability to reproduce sexually.

## Results

### Genome-wide polymorphism analyses of global *C. parasitica* isolates

We analyzed complete genomes of 230 *C. parasitica* isolates covering the global chestnut blight distribution range, as well as the European outbreak region of the invasive S12 lineage (*Figure 1A*, *Supplementary file 1*). Isolates were sequenced at a mean depth of 8–53× to detect high-confidence genome-wide SNPs. A region of 179,501–2,084,312 bp on scaffold 2 was associated to the mating type locus based on association mapping p-values (*Figure 1—figure supplement 1*). This region encoded known mating type-associated genes and was characterized by a high SNP density

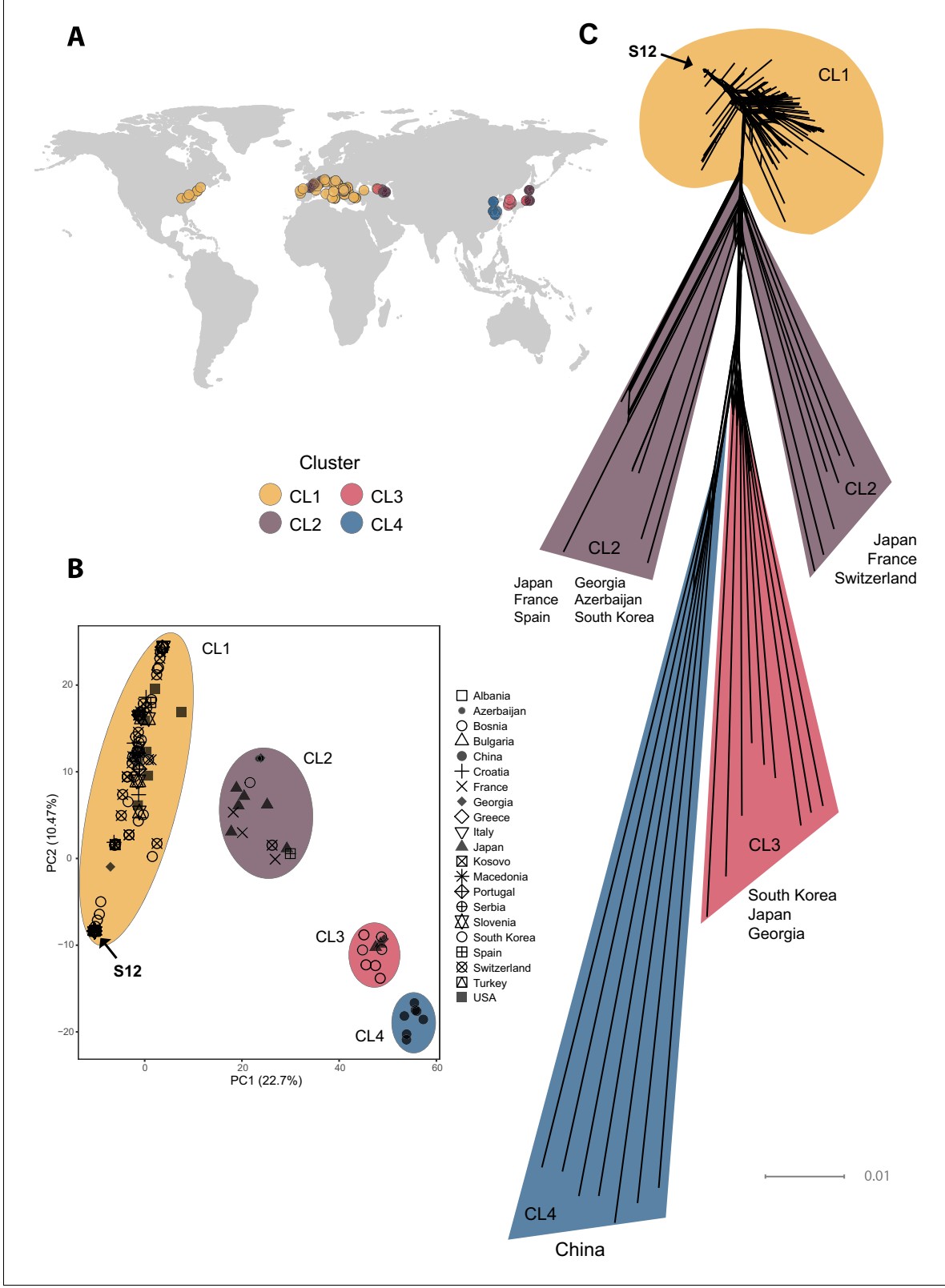

**Figure 1.** Genome-wide analyses of global *Cryphonectria parasitica* isolates. (A) Map of global sampling locations and (B) principal component analysis (PCA) of the 230 sequenced *C. parasitica* isolates. Colors indicate PCA clusters (CL1–CL4) and are as in (A). (C) SplitsTree phylogenetic network of the global *C. parasitica* sample set. Colors are as in (A) and (B). Isolates belonging to the S12 lineage are marked with an arrow in (B) and (C).

*Figure 1 continued on next page*

*Figure 1 continued*

The online version of this article includes the following figure supplement(s) for figure 1:

**Figure supplement 1.** Genome-wide association mapping (GWAS) outcome for the mating type (MAT) locus.

**Figure supplement 2.** Distribution of observed single nucleotide polymorphism (SNP) filtration quality values.

consistent with observations in other fungi (*Idnurm et al., 2015*; *Taylor et al., 2015*). We removed SNPs within the mating type-associated region to avoid confounding genetic structure with mating type divergence. We retained 80,700 SNPs and performed a principal component analysis (PCA) and constructed a SplitsTree phylogenetic network (*Figure 1B, C*). The PCA revealed four major clusters (CL1–CL4, *Figure 1B*), showing weak geographic clustering except for isolates of Chinese origin (cluster CL4 in *Figure 1B*). The largest cluster contained the majority of sequenced European isolates, including the S12 lineage, as well as all North American and two Georgian isolates (cluster CL1 in *Figure 1B*). The phylogenetic network analysis revealed large phylogenetic distances outside the main European/North American CL1 cluster (*Figure 1B, C*).

## Reconstruction of the European demographic history

We implemented a demographic model based on each cluster identified by the PCA (*Figure 1B*), merging isolates from the two smaller clusters (i.e., CL3 and CL4, *Figure 1B*) to obtain a sufficient sample size. The folded site frequency spectrum (SFS) for each of the three groups contained a large fraction of singletons, suggesting a population expansion (*Figure 2*). We used the software *dadi* (*Gutenkunst et al., 2009*) to identify the best fitting model considering no population size change or a single instantaneous population size change, as well as more complex models represented in *Figure 2* (*Supplementary file 2*). For the isolates outside the main European/North American group, we found support for a recent population expansion (CL2–CL4 clusters, *Figure 1B*, *Figure 2*). Indeed, for both clusters, the best Akaike information criterion (AIC) value was obtained for modeling a simple instantaneous expansion (AIC of 70.4 and 115.4 for the CL3+CL4 and CL2, respectively; *Supplementary file 2*). The second-best model was a more complex version of an expansion scenario including an initial bottleneck followed by exponential growth (AIC of 75.0 and 152.7 for the 'bottlegrowth' model, for the CL3+CL4 and CL2, respectively). However, for the large European/North American CL1 cluster, despite removing all but a single isolate of the S12 lineage, parameter optimization was not successful for any model or any randomly drawn initial parameters (see *Supplementary file 2*). This is likely explained by the unusual frequency spectra. As shown in *Figure 2*, the observed folded SFS for the European/North American cluster (CL1 *Figure 1B*, *Figure 2*) is W-shaped (red line in *Figure 2*) instead of a monotonously declining curve as is the case in the empirical spectra for other populations as well as in the simulations (*Figure 2*). The unusual frequency spectrum of the European/North American cluster potentially stems from the presence of multiple small groups of related genotypes due to inbreeding.

## Phylogenomic analyses of European and North American isolates

To further investigate the European/North American CL1 cluster containing the S12 lineage (*Figure 1B*), we performed a SplitsTree phylogenetic network analysis to account for reticulation caused by recombination in this cluster specifically. The network showed a substantial diversification, with both long branching and reticulation (*Figure 3A*, *Figure 3—figure supplement 1*). Genotypes within the cluster show significant evidence for recombination (pairwise homoplasy index [PHI] test; p<0.0001, *Figure 3—figure supplement 2*). Despite the high level of genetic diversity, we found no evidence for geographic structure. Moreover, we found no clustering of isolates belonging to the same vegetative compatibility type with the exception of some EU-01, EU-02, and EU-12 (S12) genotypes of predominantly Balkan origin (*Figure 3A*, *Figure 3—figure supplement 1*). Vegetative compatibility-type diversity has been widely used as a proxy for genetic diversity in populations (*Heiniger and Rigling, 1994*). Our findings confirm the clear limitation of these markers as proxies for diversity (*Ježić et al., 2012*). Nevertheless, vegetative compatibility type diversity impacts mycovirus transmission dynamics and remains, hence, a relevant population trait. Nearly all *C. parasitica* isolates representing the S12 lineage showed almost identical genotypes and tight clustering. The most tightly clustered S12 genotypes were all of mating type MAT-1 (*n* = 105). Consistent with

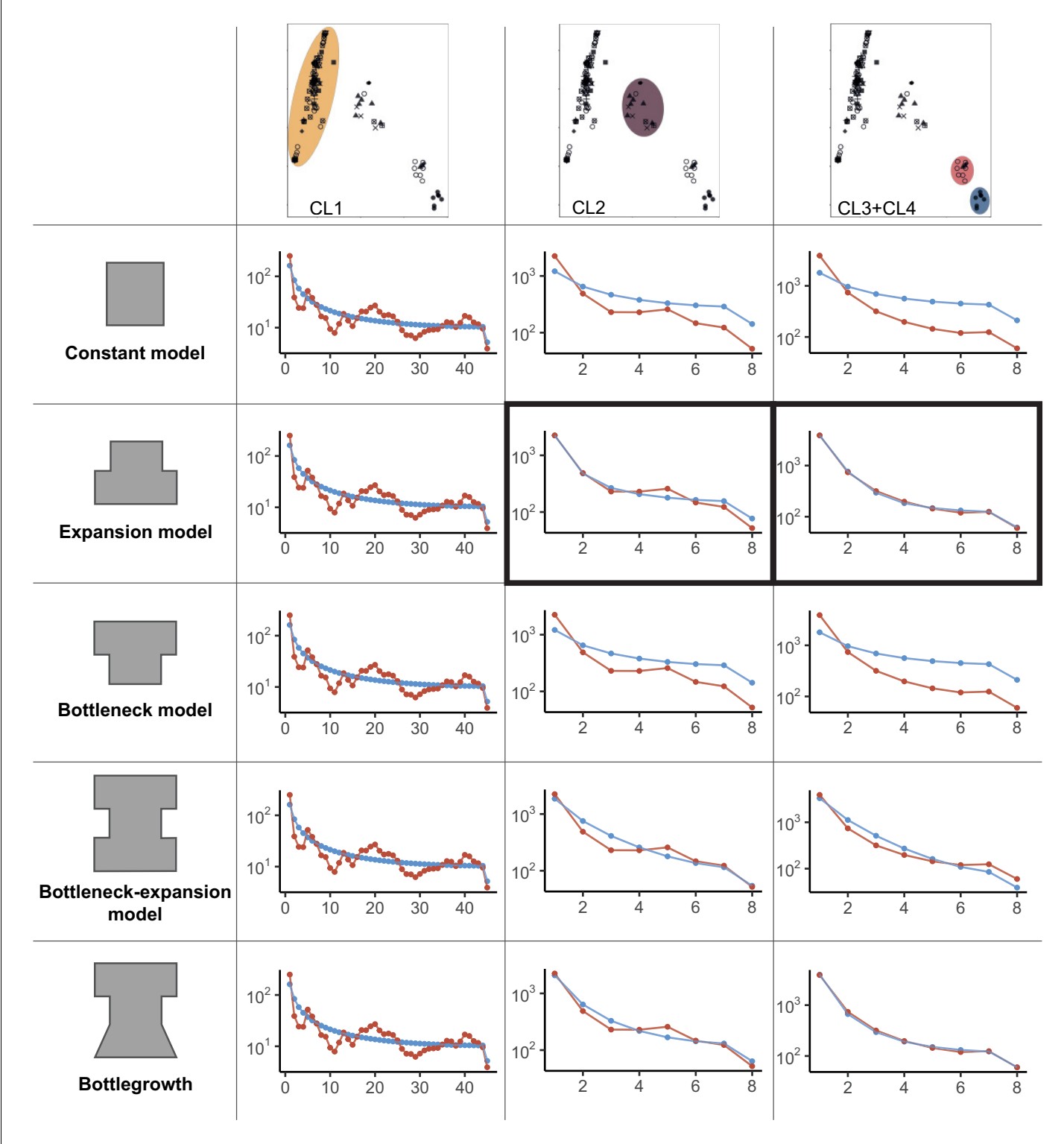

**Figure 2.** Demographic models for global *Cryphonectria parasitica* populations. Constant, expansion, bottleneck, bottleneck-expansion, and bottlegrowth population models were tested for three clusters (i.e., European/North American CL1, mixed European/Asian CL2, and Asian CL3 and CL4 clusters shown by color highlights). Clusters were inferred by principal component analysis (PCA) (see *Figure 1B*). The two smaller Asian clusters (top-right PCA panel) highlighted in red and blue were combined for demographic analyses to obtain a sufficient sample size. Panels in columns 2–4 show model—data comparison plots for the three tested scenarios (i.e., constant, expansion and bottleneck, bottleneck-expansion and bottlegrowth) as inferred by *dadi*. Blue data points and line represent the model, and red lines show the empirical data. Best fitting models according to the Akaike

*Figure 2 continued on next page*

*Figure 2 continued*
information criterion (AIC) model comparisons are marked in black boxes. A detailed list of initial and best-fit parameters, as well as likelihood and AIC values, is shown in *Supplementary file 2*.

analyses by *Milgroom et al., 2008*, this group represents the invasive S12 lineage at the origin of the expansion of *C. parasitica* across southeastern Europe. Additionally, the phylogenetic network revealed closely related but not identical S12 genotypes of mating type MAT-2 (*n* = 7, *Figure 3A*). Hence, S12 outbreak strains of MAT-2 connect the nearly uniform cluster of S12 MAT-1 strains with the remaining genetic diversity of the major European subgroup of *C. parasitica*. The S12 cluster was furthermore connected with the remaining genotypes of the major clade by two EU-12 isolates from Bosnia (M1808 with MAT-1) and Georgia (MAK23 with MAT-2) (*Figure 3A*, *Figure 3—figure supplement 1*). Consistent with previous findings (*Milgroom et al., 2008*), we also found non-S12 genotypes in the invasion zone of the S12 outbreak including the vegetative compatibility types EU-01 and EU-02 in southern Italy, Greece, and Turkey (*Figure 3A, B*). Additionally, we found the S12 lineage in otherwise more diverse regions of Croatia and Bosnia (*Figure 3A, B*). Surprisingly, we also detected one Georgian (GEZ45) and one Portuguese (M2135) isolate within the S12 cluster, suggesting human-mediated dissemination of the lineage outside of southeastern Europe (*Figure 3B*).

## Evidence for selection in European populations

We investigated potential selective sweeps in the focal European/North American cluster. To avoid a bias by the deep sampling of the S12 lineage, we excluded all but one of the S12 MAT-1 isolates (see 'Materials and methods'). We used RAiSD (*Alachiotis and Pavlidis, 2018*) to produce a composite score of selective sweep signals and identified two strong outlier loci. Despite no clear demographic model fitting the genomic data of the focal population, we nevertheless attempted to control for potential demographic history effects. We performed 10,000 simulations of a population under three different models without selection: no population size change (i.e., constant), expansion, and bottleneck. This was followed by a genome-wide selection scan on the simulated datasets using RAiSD. We assessed above what μ values selection signals are most likely exceeding effects by demographic events. The highest values of μ were obtained for the simulations with the expansion scenarios (μ = 37). Consequently, we used this value as a conservative threshold to identify the most robust signatures of selection. Two outlier loci obtained in the selection scan on the observed data have μ values above 37. The strongest sweep locus was located at the boundary of the mating type locus on scaffold 2 (*Figure 3C*). The second sweep locus was detected on scaffold 1 and encompassed an ~120 kb locus at positions 2099–3309 kb. The region contains 335 genes of which 107 encode conserved protein domains (*Figure 3C*, *Supplementary file 3*). Hence, genetic variation in the two selection sweep regions is potentially underpinning recent adaptation of *C. parasitica* to the European environment.

## Analyses of coancestry for the invasive lineage

The SplitsTree network revealed that the invasive S12 lineage has closely related genotypes occurring in Europe. Thus, to dissect the genetic ancestry of S12, we performed a coancestry matrix analysis using fineSTRUCTURE (*Lawson et al., 2012*) considering all isolates of the major European/North American cluster (CL1, *Figure 1B*), including the S12 lineage (*n* = 197). Within this cluster, K = 43 genetic subclusters were identified with fineSTRUCTURE (*Figure 4*, *Figure 4—figure supplement 1A*). The large number of identified clusters likely stems from the presence of multiple small groups of highly related individuals beyond the S12 lineage. Furthermore, populations only showed weak geographic structuring (*Figure 4—figure supplement 2*). The highest level of coancestry for any given S12 isolate was found within the S12 population (*Figure 4*). Moreover, we identified the highest degree of shared ancestry between S12 and populations from the northern Balkans (Bosnia and Croatia), southern Switzerland, but also Georgia and Spain, with coancestry values between 35.3 and 49.7 (*Figure 4*, *Figure 4—figure supplement 2*). As Central European *C. parasitica* populations were likely established from North American sources alone (*Dutech et al., 2012*), the coancestry analysis points to a European origin of the invasive S12 lineage rather than a novel introduction from North America.

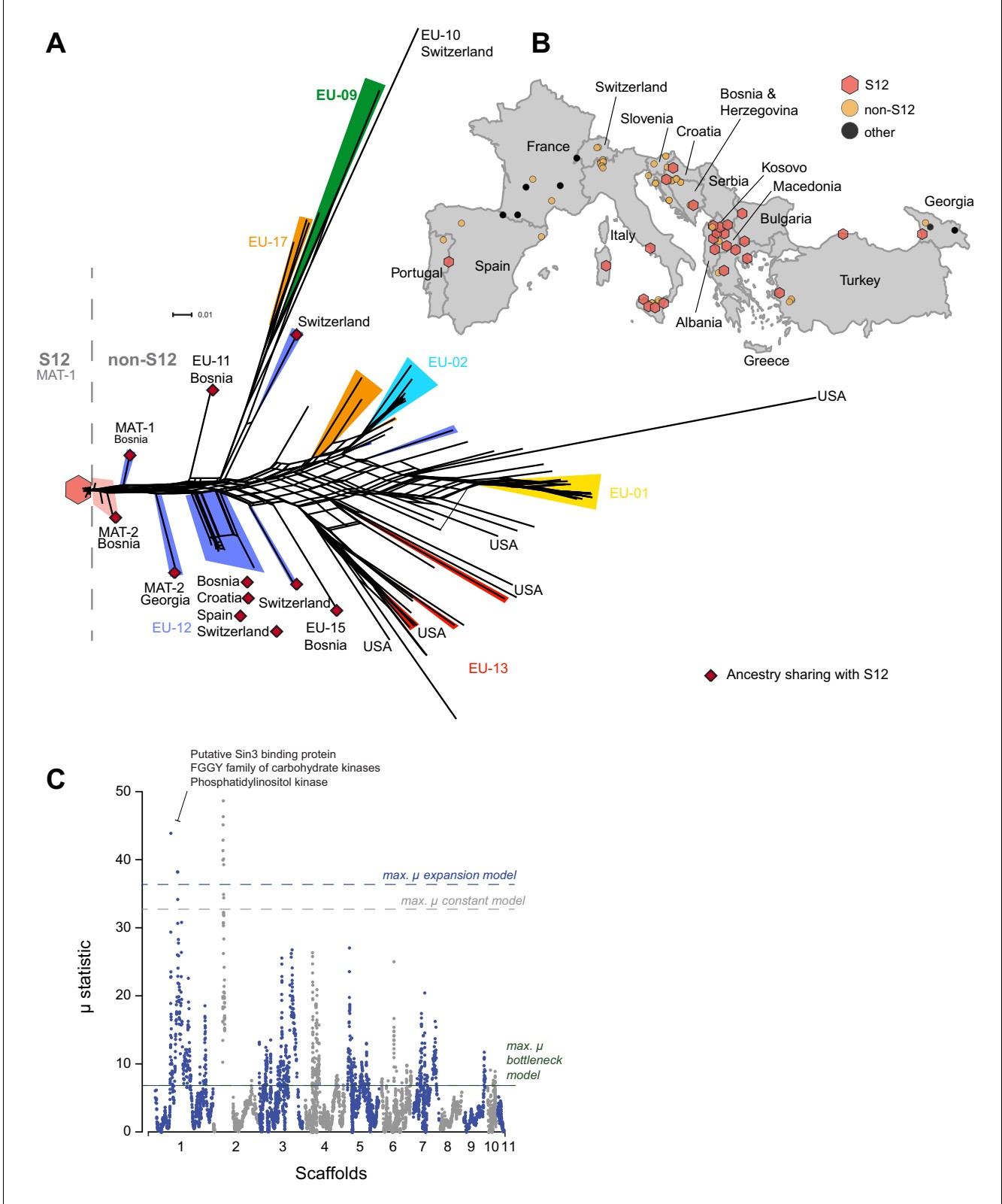

**Figure 3.** Phylogenetic network structure of the main European/North American subgroup and spatial distribution of S12 across Europe. (**A**) The highlighted branches represent the most abundant vegetative compatibility types. Isolates belonging to the S12 outbreak lineage (EU-12; mating type MAT-1, *n* = 105) are marked with a red hexagon and match symbols in (**B**). S12 isolates of mating type MAT-2 are highlighted in light red. Additional EU-12 isolates not belonging to the S12 lineage are highlighted in blue with information on the country of origin. Isolates sharing ancestry with the S12

*Figure 3 continued on next page*

*Figure 3 continued*

lineage as inferred by fineSTRUCTURE (*Figure 4*, *Figure 4—figure supplement 2*) are marked with dark red diamonds. (B) Geographic map of European sampling locations. Light red hexagons mark locations where S12 was found, and yellow circles indicate locations containing other genotypes from the main European/North American CL1 cluster (*Figure 1B*). Black circles show the location of highly distinct genotypes outside of the CL1 cluster (*Figure 1A, B*). (C) Genome-wide scan for selective sweeps (RAiSD). Effects due to the unknown demographic history of the European/North American CL1 cluster (*Figure 1B*) were mitigated by implementing population simulations under constant, expansion, and bottleneck scenarios. The three dashed lines show the maximum values observed in the simulated datasets following constant (gray) expansion (blue) and bottleneck (green) demographic models. Encoded protein functions overlapping with the top regions are shown as summaries.

The online version of this article includes the following figure supplement(s) for figure 3:

**Figure supplement 1.** SplitsTree of all non-S12 isolates showing full isolate identifiers.

**Figure supplement 2.** Per site population recombination rate ρ of European/North American *Cryphonectria parasitica* populations as inferred with LDhat.

## TE landscape and CNV in the S12 lineage

Invasive pathogen lineages may have undergone crucial genomic rearrangements producing more fit genotypes. Here, we generated a de novo identification and annotation of TEs for the *C. parasitica* genome. We found that 12% of the genome was composed of TEs with striking variation along the assembled scaffolds (i.e., quasi-chromosomes; *Figure 5A*). In particular, regions on scaffold 2 matching the mating type locus are highly enriched in TEs, suggesting that the large non-recombining region has undergone substantial degeneration (*Figure 5A*). In contrast, the vegetative incompatibility (*vic*) loci are located in regions devoid of TEs. In fungal pathogens, effector genes and TEs are often co-localized in fast-evolving compartments of the so-called 'two-speed genome' (*Dong et al., 2015*). However, the *C. parasitica* genome shows no apparent compartmentalization into gene-sparse and gene-rich regions. We used machine learning to predict secreted proteins most likely acting as effectors to manipulate the host. In contrast to other plant pathogens, effector gene candidates showed no tendency to localize in gene-sparse regions of the genome (*Figure 5B*). Non-repressed TEs can potentially create additional copies in the genome, leading to intra-species variability in TE content. To detect such TE activity, we performed genome-wide scans of *C. parasitica* isolates for presence or absence of TEs based on split read and target site duplication information. At loci with detectable TE presence/absence polymorphism, we found an over twofold variation in total TE counts across all isolates (*Figure 5C*). The total TE count variation among the genetically diverse non-S12 isolates (North America and Europe only, *Figure 3A*) was larger than the clonal S12. Nevertheless, the count of TE insertions at polymorphic TE insertion sites ranged between 12 and 52 among S12 isolates, which was surprisingly high given their recent emergence and extremely high similarity across the genome (*Figure 5C*). Across TE insertion loci, we found only a marginally significant difference in the TE frequency among S12 compared to non-S12 isolates ($p=0.03$, *Figure 5—figure supplement 1*). This is in strong contrast to segregating SNPs. We found pairwise differences of 0–45 SNPs among S12 isolates and 0–3770 among non-S12 isolates (test statistic t-test mean difference $p<0.001$; *Figure 5—figure supplement 2*). For a direct comparison of TE and SNP diversity, we assessed the coefficient of variation (CV) of normalized TE and SNP counts. For TEs, we found a $CV = 0.24$ for S12 and $CV = 0.33$ for non-S12 isolates. For SNPs, we found a $CV = 0.007$ for S12 and $CV = 0.69$ for non-S12. Hence, our findings indicate a higher TE diversity among isolates compared to SNP diversity within the S12 lineage. We found that 44% and 49% of the polymorphic TE insertion sites are specific to the S12 and non-S12 group of isolates, respectively. Given the much more recent origin of the S12 lineage, this suggests increased TE activity since the emergence of the S12 lineage contributing to the creation of de novo genetic variation. For SNPs but not for TEs, the minor allele frequency spectrum of the S12 lineage was significantly skewed toward low-frequency variants compared to non-S12 isolates (Fisher's exact test, $p<0.001$; *Figure 5—figure supplement 2*).

Repetitive sequences such as TEs can trigger non-homologous recombination leading to CNV including deletions. We used normalized read coverage to assess CNV across the species including the S12 lineage (*Figure 5A*). Compared to the reference genome, we found that most CNVs in the S12 lineage are nearly fixed. Non-S12 isolates segregate most CNVs at intermediate frequencies (comparison between S12 and non-S12; Fisher's exact test $p<0.001$, *Figure 5—figure supplement 1*). Genes tend to overlap duplications rather than deletions (odds ratios of 0.37 and 0.13,

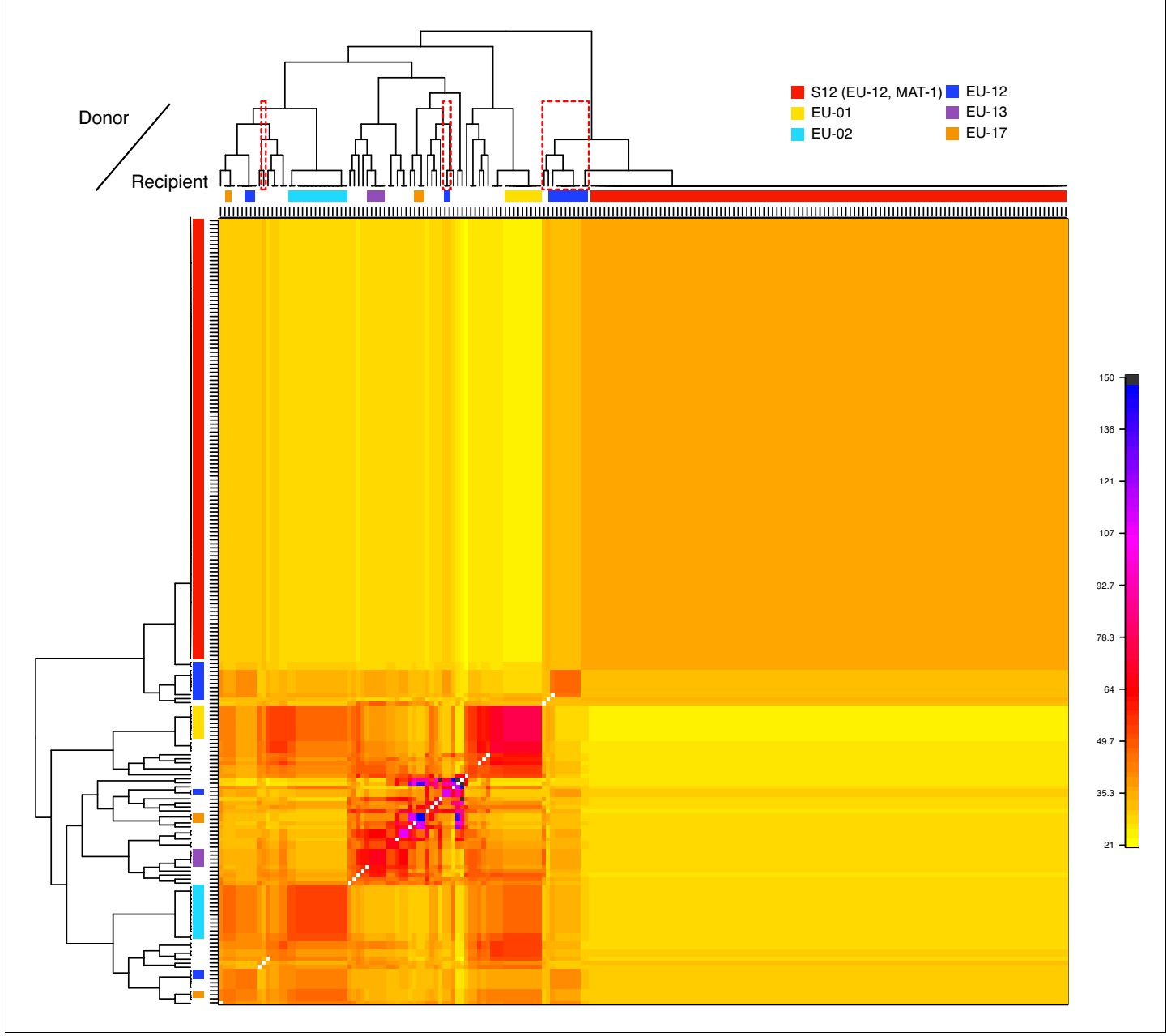

**Figure 4.** Analysis of donors to the S12 lineage genotypes. Averaged coancestry matrix and population tree of the North American/European subgroup estimated by fineSTRUCTURE. The heatmap indicates averaged coancestry between populations. Populations sharing ancestry with S12 are marked in red-dashed boxes. S12 ancestry-sharing populations are additionally marked in *Figure 3A*. Detailed coancestry matrix with extensive population information is shown in *Figure 4—figure supplement 2*.

The online version of this article includes the following figure supplement(s) for figure 4:

**Figure supplement 1.** fineSTRUCTURE convergence plots.

**Figure supplement 2.** Full averaged coancestry matrix showing isolate identifiers.

respectively; two-sided Fisher test, p-value<0.001). TEs tend to overlap deletions rather than duplications (odds ratios of 5.03 and 1.76, respectively; two-sided Fisher's test, p-value<0.001) (*Figure 5D*). The mating type region on scaffold 2 and the rDNA locus on scaffold 6 show particularly high levels of CNV (*Figure 5A*). In a joint analysis of all isolates from the main European/North American cluster (CL1, *Figure 1B*), we found that coding sequences overlapping with duplications and deletions are enriched for gene ontology terms associated with protein binding functions and

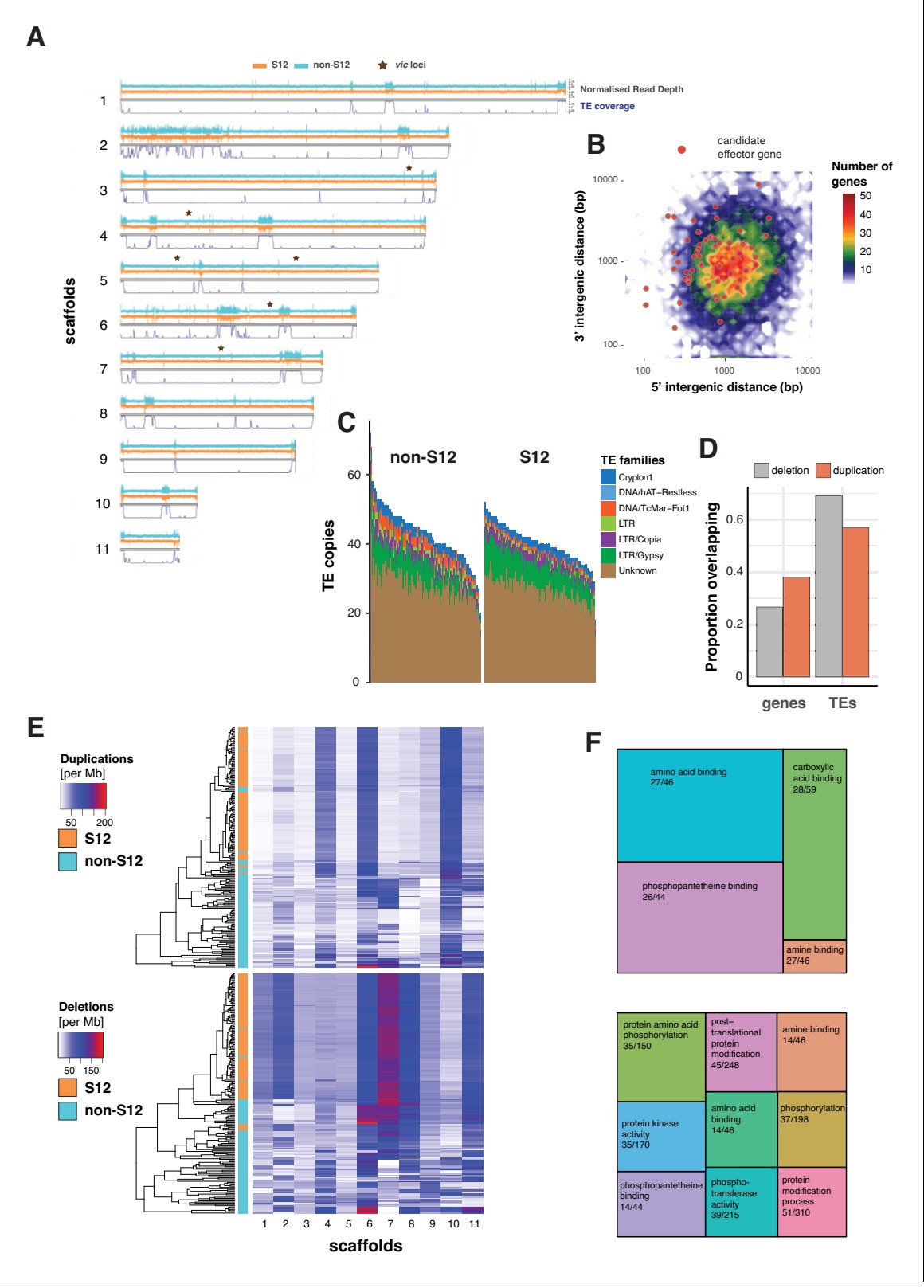

**Figure 5.** Transposable element (TE) landscape and copy number variation. (A) Genome-wide coverage of transposable elements (in 10 kb windows) matched by with normalized read depth for the S12 and non-S12 lineages (North America and Europe only). *vic* loci: vegetative incompatibility loci. (B) Genome-wide distribution of intergenic distances according to the length of 5′ and 3′ flanking regions. Red dots represent genes encoding predicted effector proteins. (C) Counts of detected TE sequences across S12 and non-S12 isolates using split reads and target site duplication information. (D)

*Figure 5 continued*

Proportion of normalized read depth windows (800 bp) with evidence for duplications (normalized read depth >1.6) or deletions (<0.4) overlapping with genes and TEs. (E) Heatmap showing the number of windows (800 bp) with duplications and deletions. The dendrogram shows the similarity in duplication or deletion profiles for S12 and non-S12 isolates (North America and Europe only). (F) Molecular functions (based on gene ontology) enriched in duplicated and deleted regions (upper and lower panel, respectively). Enrichment was tested by hypergeometric tests, and significance was established for a Bonferroni threshold at alpha = 0.05. The numbers represent the number of genes with the matching gene ontology term in a duplicated or deleted region, and across the genome, respectively.

The online version of this article includes the following figure supplement(s) for figure 5:

**Figure supplement 1.** Genetic diversity within the S12 lineage.
**Figure supplement 2.** Pairwise genome-wide nucleotide differences between S12 and non-S12 isolates.
**Figure supplement 3.** Normalized read depth distribution across C. parasitica isolates.

protein phosphorylation activity, respectively (*Figure 5F*). Overall, our results show that the S12 lineage underwent specific gene deletion and duplication patterns compared to the broad diversity of non-S12 isolates.

## The evolutionary history of European populations

To gain insights into evolutionary forces shaping polymorphism in the outbreak S12 mating type MAT-1 lineage versus other populations in the European/North American cluster (non-S12, CL1 cluster; *Figure 1B*, *Figure 3A*), we first analyzed allele frequencies across the genome in both groups (*Figure 6A*). The S12 lineage segregated virtually no intermediate allele frequencies in the range of 0.05–0.95. In contrast, the non-S12 genotypes showed overall a wide spectrum of allele frequencies across the genome. Genome-wide nucleotide diversity was extremely reduced in the S12 compared to non-S12 populations (*Figure 6B*). We analyzed the predicted impact on protein functions of segregating mutations in the S12 lineage and non-S12 populations (*Figure 6C*). We found 4 highly and 104 moderately deleterious mutations segregating within S12 compared to 32 high and 972 moderately deleterious mutations in non-S12 groups (*Figure 6C*). Three of the high-impact SNPs in the S12 lineage were classified as stop gain mutations, as well as one splice acceptor variant (insertion variant). Two of these high-impact mutations affect proteins of the major facilitator superfamily, as well as a protein containing a LCCL domain and an ecdysteroid kinase. Non-S12 populations showed an over-representation of low-frequency high-impact mutations (Fisher contingency table test odds ratio of 3.7 and p=0.002028; *Figure 6C*). This is consistent with purifying selection reducing the frequency of these mutations due to fitness costs. Within the S12 lineage nearly all segregating mutations were at very low frequency. We found only modifier (i.e., nearly neutral) mutations rising to higher frequency within the lineage, suggesting that deleterious mutations in the S12 lineage can still be removed through low levels of recombination and purifying selection. The extremely low level of polymorphism segregating within the S12 lineage prevents strong inferences of selective sweeps in the lineage.

## Retracing S12 invasion routes

To infer potential invasion routes of the S12 outbreak lineage (*n* = 105; *Figure 3A*), we investigated intra-lineage genetic diversity across southeastern Europe. The closely related genotypes segregated 529 high-confidence SNPs across the genome, of which 459 were identified as singletons. The phylogenetic network revealed a star-like structure, consistent with a predominantly clonal population structure (PHI test p>0.05; *Figure 7A*). The genetic structure assessed by a PCA suggests a differentiation into two to three groups with a weak geographic association (*Figure 7B*). The majority of isolates were found in a cluster along the principal component axis 2 (*Figure 7B*) without apparent geographic structure within the group. The second group consisted of eight isolates from Turkey and one isolate from Georgia consistent with the geographic proximity of the countries (*Figure 7B*). The most diverse group consisted exclusively of isolates from Greece showing reticulation in the SplitsTree (*Figure 7A*). Nucleotide diversity for the S12 outbreak lineage was $\pi = 2.90 \times 10^{-7}$ compared to $\pi = 2.62 \times 10^{-7}$ for the largest subgroup (*n* = 89) and $\pi = 1.80 \times 10^{-7}$ for the Turkish/Georgian subgroup (*n* = 8) excluding the diverse group of Greek isolates. The spread of the Greek isolates suggests that these might have recently undergone sexual reproduction and admixture

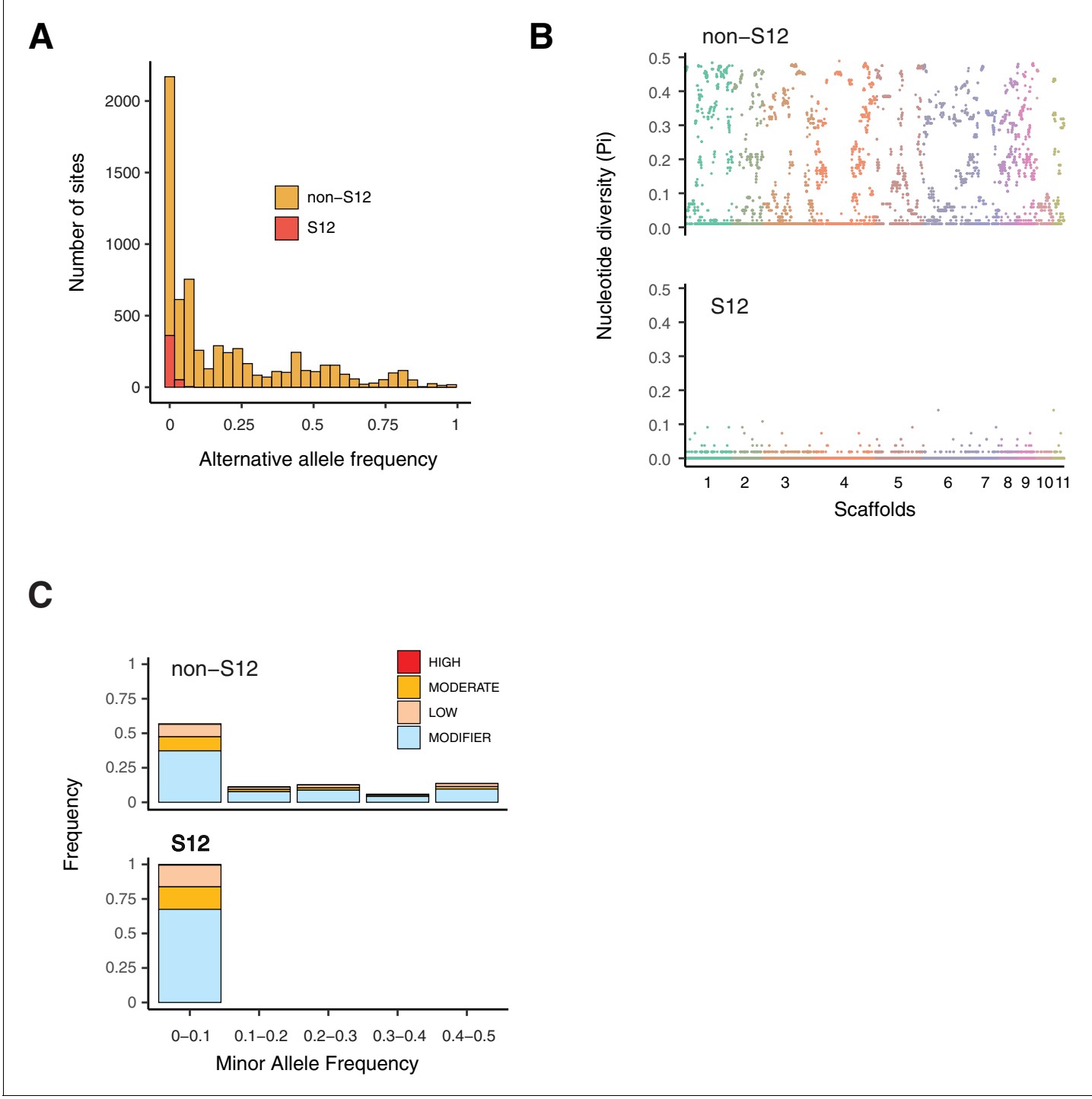

**Figure 6.** Polymorphism segregating in the S12 lineage. (**A**) Alternative allele frequencies spectra across the genome for the S12 lineage (*n* = 105) compared to all other analyzed European (non-S12; *n* = 92). (**B**) Genome-wide nucleotide diversity (Pi) for the S12 lineage and non-S12 lineages in 10 kb windows. (**C**) Minor allele frequency spectra of high, moderate, and modifier (i.e., near neutral) impact mutations as identified by SnpEff.

events. As we identified non-S12 genotypes in Greece, either within-lineage recombination or exchange with other lineages may underlie the reticulation and increased diversity.

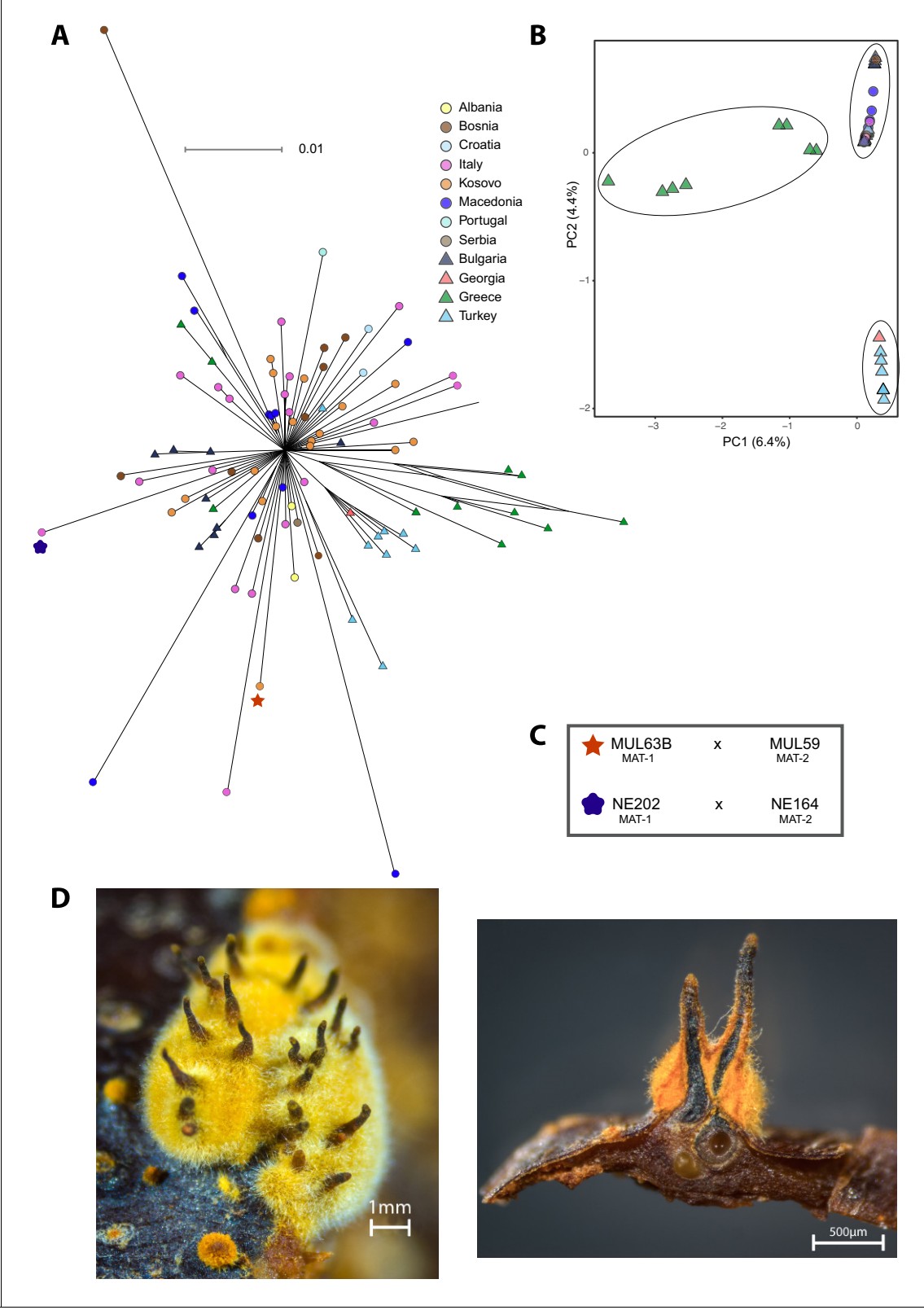

**Figure 7.** Fine-scale genetic diversity analyses of the S12 lineage. (**A**) SplitsTree and (**B**) principal component analysis (PCA) of the S12 mating type MAT-1 outbreak isolates (*n* = 105). Symbols and colors are as in (**A**). (**C**) Scheme of successful mating pairs of S12 mating type MAT-1 isolates crossed with isolates from the opposite mating type of the same geographic origin. Symbols are as in (**A**). (**D**) Photographic images of sexual *C. parasitica* fruiting bodies (i.e., perithecia) emerging from crosses of S12 mating type MAT-1 isolates with isolates of the opposite mating type after 5 months of

*Figure 7 continued on next page*

*Figure 7 continued*

incubation under controlled conditions. Left: perithecia embedded in a yellow-orange stromatic tissue. Right: cross-section of perithecia and chestnut bark. Flask-shaped structures with a long cylindrical neck develop in yellow-orange stromatic tissue and are embedded in the bark (except for the upper part). The ascospores are formed in sac-like structures (asci) in the basal part of the perithecium. When mature, the ascospores are actively ejected into the air through a small opening (ostiole) at the end of the perithecial neck.

## Experimental tests of S12 mating competence

We tested experimentally whether S12 mating type MAT-1 isolates were still able to reproduce sexually. We selected S12 isolates from populations in Bosnia, Kosovo, and Italy matching locations where the opposite mating type was also detected (*Supplementary file 4*). After 5 months of incubation, we confirmed outcrossing of isolates of opposite mating type within the S12 lineage by pairing isolates from Molliq (Kosovo) and Nebrodi (southern Italy) (*Figure 7C*). Mating pairs from Molliq and Nebrodi grew numerous perithecia, which are the fruiting bodies specific to sexual reproduction (*Figure 7D*). Pairings of Bosnian isolates showed no perithecia formation. This could have been caused by undetected variation in isolate viability affecting mating competence, mutation accumulation, or environmental sensitivity. Using molecular mating type assays, we recovered both mating types among the ascospores produced from successful matings.

## Discussion

We analyzed the invasion of the chestnut blight pathogen *C. parasitica* across southeastern Europe after a first establishment on the continent. The European population is characterized by the presence of a small number of highly distinct genotypes. Deep sampling of recent invasive genotypes showed that the secondary invasion in southeastern Europe was caused by a single, highly homogeneous lineage consisting nearly exclusively of a single mating type. Given the likely origin from the European gene pool with clear evidence for sexual reproduction, the invasive lineage likely transitioned from biparental mating populations to a single mating type. The spread across southeastern Europe left no clear geographic imprint. This lack of genetic differentiation along the invasion route may be due to high levels of gene flow after the initial colonization or a very rapid spread from the center of origin to multiple locations across southeastern Europe. We show that ongoing TE activity created unexpected levels of insertion polymorphism within the invasive lineage. In addition, the lineage carries a distinct gene deletion and duplication profile compared to the European and North American pool of *C. parasitica.*

### The establishment of a European bridgehead population

Central and southeastern European *C. parasitica* populations likely originated from North American lineages. Clusters of European and North American genotypes were largely overlapping. Shared genotype clusters between the continents are consistent with historic gene flow from North America to Europe. These genome-scale analyses are consistent with previous findings documenting multiple North American introductions into Central Europe, but also directly from Asia to Western Europe (*Demené et al., 2019*; *Dutech et al., 2012*; *Milgroom et al., 1996*). Our results show that Central and southeastern European *C. parasitica* populations were largely established from North American sources alone consistent with previous studies. We were unable to sufficiently sample the North American source populations, and, hence, demographic history analyses of North American and European populations remained inconclusive. However, the high diversity at the genetic level with polymorphism at the level of SNPs, TEs, and CNV, as well as at the level of vegetative compatibility types, suggests that Europe was repeatedly colonized over the past century. Although genetic diversity could also have accumulated in situ in Europe, the establishment of a large set of genotypes and vegetative compatibility types seems dicult to explain with population age alone. Sexual recombination between the three most common vegetative compatibility types in Europe (i.e., EU-01, EU-02, and EU-05; *Robin and Heiniger, 2001*) could not account for the observed vegetative compatibility-type diversity. Observational records date the first introductions into Europe to the 1930s (*Robin and Heiniger, 2001*). Hence, based on the observed genetic diversity, a scenario of repeated introductions of different vegetative compatibility types since the 1930s seems most plausible.

Within Europe, populations from southern Switzerland, Slovenia, Croatia, and Bosnia are highly diverse and have nearly balanced mating type ratios. We also found no clear genetic structure according to vegetative compatibility types. This strongly suggests frequent outcrossing and population admixture, consistent with reports of perithecia in the field (*Ježić et al., 2012*; *Prospero et al., 2006*; *Prospero and Rigling, 2012*; *Trestic et al., 2001*). Low vegetative compatibility-type diversity in most European *C. parasitica* populations was thought to have contributed to low population admixture within Europe compared to Asia and North America (*Cortesi et al., 1996*; *Dutech et al., 2012*; *Prospero and Rigling, 2012*). However, our genome-wide analyses revealed frequent and ongoing in situ admixture in Europe. Thus, vegetative compatibility-type diversity does not necessarily underpin population admixture frequency and genetic diversity in sexually recombining populations. Our findings show that in asexually reproducing populations, such as in the S12 lineage, genotypes tend to cluster according to vegetative compatibility types.

## Emergence of an invasive lineage from a European bridgehead

The invasive lineage S12 most likely arose from existing genotypes established in Europe. The closest genotypes to the dominant S12 MAT-1 were S12 MAT-2 isolates found in Bosnia, Kosovo, and southern Italy. Analyses based on a coancestry matrix identified populations from Bosnia, Croatia, Switzerland, and Georgia, but also Spain sharing the highest ancestry with the S12 lineage. Although we cannot exclude the existence of yet undescribed genetic diversity in North American populations, our results strongly indicate that introductions from outside of Europe are unlikely to explain the emergence of S12. The lineage carries a unique set of copy number variants compared to other European genotypes underlining the observation of a recombinant S12 genotype. Furthermore, the emergence of S12 was accompanied by a striking evolutionary transition from mixed mating type populations to single mating type outbreak populations. Human activity may have contributed to the shift toward single mating type populations. Shipments of infected chestnut seedlings from Northern Italy and other trading activities could have disseminated the invasive lineage further South. This would have exposed the pathogen to the geographically more fragmented chestnut forests typically found in southeastern Europe where asexuality or selfing may be advantageous in the absence of mating partners. Although *C. parasitica* is able to produce asexual conidia in large quantities, these specific spores are thought to be mainly splash dispersed by rain over short distances (*Griffin, 1986*). Accounting for occasional dispersal by birds or insects (*Heald and Studhalter, 1914*), conidia dispersal is unlikely to contribute substantially to the colonization of new areas. Moreover, human-mediated dispersal could have introduced the lineage to other regions including the Iberian Peninsula and Georgia.

Despite the loss of a mating type in the S12 lineage, we found limited evidence for reticulation indicating at least low levels of recombination. If mating in *C. parasitica* follows the canonical process found in many ascomycetes, isolates of opposite mating type are required. Hence, S12 isolates of mating type MAT-1 may sporadically mate with rare S12 isolates of mating type MAT-2, which are comparatively more diverse. The emergence of the opposite mating type at low frequency could be the result of recombination with other genotypes and subsequent backcrossing. Combined with experimental evidence, we show that the dominant S12 mating type MAT-1 has retained the ability for sexual reproduction. Furthermore, in Bosnia, Croatia, Italy (Sicily), and Turkey, the S12 lineage coexists with other genotypes (i.e., vegetative compatibility types EU-01 and EU-02) of both mating types, potentially enabling sexual recombination and diversification in situ. The invasive S12 lineage was potentially pre-adapted to the southeastern European niche as we traced the origins to a likely bridgehead population located in southern Switzerland, northern Italy, or the northern Balkans. Niche availability and benefits associated with asexual reproduction to colonize new areas may have pre-disposed the European *C. parasitica* bridgehead population to produce a highly invasive lineage. Such potential pre-adaptation to the southeastern European environmental conditions, as well as reduced niche availability, may also explain why genotypes belonging to the S12 lineage have rarely been found elsewhere in Europe.

## Expansion and mutation accumulation within the invasive lineage

The S12 lineage diversified largely through mutation accumulation as nearly all high-confidence SNPs were identified as singletons. Mutation accumulation in the absence of substantial

recombination resulted in star-like phylogenetic relationships. We found a surprisingly high degree of differentiation among S12 genomes at the level of TE insertion polymorphism. Hence, active transposition of TEs is an important factor in diversifying the invasive lineage and possibly underpin future adaptive evolution. We found no evidence for a preferential association of candidate effector genes with TEs. Even though genome compartmentalization is a feature in many fungal genomes, the association of effector genes with repeat-rich regions is not consistent among pathogens (*Torres et al., 2020*). Necrotrophic pathogens like *C. parasitica* often rely less on effector-based gene-for-gene interactions and tend to deploy toxins or cell-wall degrading enzymes (*Friesen et al., 2008*; *Torres et al., 2020*). Interestingly, *C. parasitica* has recently experienced a loss of genes associated with carbohydrate metabolism and cell-wall degradation, which may have reduced exposure to the host immunity system and increased virulence (*Stauber et al., 2020*). The lack of genes with confirmed pathogenicity functions largely prevents thorough investigations on factors facilitating virulence evolution (e.g., epigenetic silencing or sequence rearrangements).

Analyses of allele frequency spectra suggested that the broader European *C. parasitica* populations effectively removed the most deleterious mutations through purifying selection. In contrast, the S12 lineage shows strong skews toward very low minor allele frequencies of all mutation categories. Interestingly, we found a broader spread in allele frequencies for nearly neutral mutations in the S12 lineages. This suggests that despite the largely clonal population structure deleterious mutations can still be removed through low levels of recombination and purifying selection. Using accumulated mutations as markers to retrace the spatial expansion of the invasive S12 lineage, we found no indication for a step-wise geographic expansion along potential invasion routes. A lack of genetic clustering across southeastern Europe may be a consequence of high levels of potentially human-mediated gene flow frequently introducing new genotypes over large distances. However, the lack of geographic structure could also have its origins from substantial population bottlenecks during the spread of S12 across southeastern Europe. Finally, the largely clonal lineage may also become exposed to processes such as Muller's Ratchet fixing deleterious mutations over time (*Felsenstein, 1974*).

We show how a highly invasive fungal pathogen lineage in Europe can emerge from an intermediate, genetically diverse bridgehead population in Europe. This is in line with the self-reinforcement invasion model where initial introductions promote secondary spread (*Bertelsmeier et al., 2018*; *Garnas et al., 2016*). However, empirical evidence for adaptation in bridgehead populations is often elusive. Additionally, human activity (*Banks et al., 2015*) and host naivety of the European chestnut could have contributed substantially to the rapid spread without the need for diversification and adaptation in the bridgehead population. To date, there is no evidence for the emergence of tolerance or resistance against *C. parasitica* in European chestnut populations. However, fungal virulence may be severely reduced by the naturally occurring or artificially introduced parasitic mycovirus CHV1 (*Rigling and Prospero, 2018*). Interestingly, mycovirus spread is most efficient in asexual *C. parasitica* populations such as the invasive S12 lineage. This is because isolates of the same lineage lack vegetative incompatibility barriers among each other. The mycovirus has been reported in southeastern Europe where it shows population structure in contrast to the fungal host (*Bryner and Rigling, 2012*). Our study included only CHV1-free S12 isolates, hence potential associations of viral load, genetic identity, and geographic structure remain to be investigated. Outcrossing populations often harbor many different vegetative compatibility groups slowing viral transmission (*Robin and Heiniger, 2001*). Hence, the presence of the mycovirus may confer a selective advantage to retain sexual cycles and diversify vegetative compatibility types. In turn, diversification may reduce the evolutionary advantage of the invasive lineage over time.

## Materials and methods

### Samples of *C. parasitica*

A total of 230 *C. parasitica* were sequenced. A majority of 182 isolates originated from Albania, Bosnia, Bulgaria, Croatia, Greece, Italy, Kosovo, Macedonia, Serbia, Slovenia, Switzerland, and Turkey (*Figure 3B*, *Supplementary file 1*). All 182 isolates originating from Central and southeastern Europe tested to be virulent and mycovirus-free. The 48 other isolates were from China ($n = 7$), South Korea ($n = 8$), Japan ($n = 8$), North America ($n = 8$), Azerbaijan ($n = 3$), Georgia ($n = 4$),

Portugal (*n* = 3), Spain (*n* = 3), and France (*n* = 5) (*Supplementary file 1*). A total of 125 European isolates belonged to the vegetative compatibility type EU-12, whereas 57 isolates represented other vegetative compatibility types (EU-types) occurring in Central and southeastern Europe. We additionally selected 45 isolates from Bulgaria, Greece, Italy, and Macedonia, which were already included in a previous population-wide study on southeastern European *C. parasitica* diversity by *Milgroom et al., 2008*. All samples were collected between 1951 and 2018 and are stored as glycerol stocks at –80°C in the culture collection of the Swiss Federal Research Institute WSL.

## DNA extraction and genotyping

All isolates were first inoculated onto cellophane-covered potato dextrose agar plates (PDA, 39 g/L; BD Becton, Dickinson and Company, Franklin Lakes, USA) (*Hoegger et al., 2000*) and incubated for a minimum of 1 week at 24°C, at a 14 hr light and 10 hr darkness cycle. After a sufficient amount of mycelium and spores had grown, the isolates were harvested by scratching the mycelial mass off the cellophane, transferring it into 2 mL tubes and freeze-drying it for 24 hr. For DNA extraction, 15–20 mg of dried material was weighted and single-tube extraction was performed using the DNeasy Plant Mini Kit (Qiagen, Hilden, Germany). DNA quantity was measured using the Invitrogen Qubit 3.0 Fluorometer (Thermo Fisher Scientific, Waltham, MA), and DNA quality was assessed using the Nanodrop 1000 Spectrophotometer (Thermo Fisher Scientific). Prior to sequencing, all isolates were screened for their genotype at 10 microsatellite markers (*Prospero and Rigling, 2012*). Additionally, the isolates were screened for their vegetative compatibility and mating type alleles in two multiplex PCRs, as described in *Cornejo et al., 2019*. Allele sizes for the genotyping of vegetative compatibility and mating types were scored with GeneMapper 5 (Thermo Fisher Scientific).

## Illumina whole-genome sequencing, variant calling, and filtration

Isolates were prepared for sequencing using the TrueSeq Nano DNA HT Library Preparation kit (Illumina, San Diego, CA). The libraries were sequenced at the Functional Genomics Centre Zurich (ETH Zurich and University of Zurich) on Illumina platforms (Illumina). The obtained sequences were trimmed with Trimmomatic v0.36 (*Bolger et al., 2014*) and aligned with Bowtie 2 v2.3.5.1 (*Langmead and Salzberg, 2012*) and SAMtools v1.9 (*Li et al., 2009*) to the *C. parasitica* reference genome (43.9 Mb) EP155 v2.0 (*Crouch et al., 2020*) available at the Joint Genome Institute (http://jgi.doe.gov/). Variant calling, selection, and filtration were conducted with the Genome Analysis Toolkit GATK v3.8 and v4.0.2.0 (*McKenna et al., 2010*). We retained variants satisfying the following filtration parameters: QUAL:$\geq$100, MQRankSum (lower):$\geq-2.0$, QD:$\geq$20.0, MQRankSum (upper):$\leq$2.0, MQ:$\geq$20.0, BaseQRankSum (lower): $\geq - 2.0$, ReadPosRankSum (lower):$\geq-2.0$, ReadPosRankSum (upper):$\leq$2.0, BaseQRankSum (upper):$\leq$2.0 (*Figure 1—figure supplement 2*). Furthermore, we used BCFtools v1.9 (*Narasimhan et al., 2016*) and the R package vcfR (*Knaus and Grünwald, 2017*) for querying and VCFtools v0.1.16 (*Danecek et al., 2011*) for downstream variant filtering. Variants were additionally filtered for minor allele count $\geq$1, excluding all missing data. Sites were also filtered per genotype, only keeping biallelic SNPs with a minimum depth of 3 and a genotyping quality of 99. To exclude SNPs associated with the mating type, we ran an association study with TASSEL 5 (*Bradbury et al., 2007*) and retrieved p-values for each SNP across the genome. We set a p-value threshold of $p\leq1\times10^{-10}$ to remove all mating type-associated SNPs for further analysis. The SNPs showing strong association with the mating type were located on scaffold 2 between positions 179,552 – 2,084,312 bp (*Figure 1—figure supplement 1*).

## Phylogenetic reconstruction

The filtered whole-genome SNP dataset was used to build unrooted phylogenetic networks using SplitsTree v4.14.6 (*Huson and Bryant, 2006*). SplitsTree was also used for calculating the PHI test (*Bruen et al., 2006*) to test for recombination. The required file conversions for SplitsTree (i.e., from VCF to FASTA format) were done with PGDSpider v2.1.1.5 (*Lischer and Excoffier, 2012*). PCA was performed as implemented in the R package ade4 (*Bougeard and Dray, 2018*).

## Inference of S12 donor populations

We generated an averaged coancestry matrix as inferred by fineSTRUCTURE v2.1.3 (*Lawson et al., 2012*). The software uses a MCMC-based algorithm to infer ancestral contributions based on

patterns of haplotype similarity. For analysis, we selected the unlinked model and generated the required input files ('phase' files) in R. We ran the fineSTRUCTURE pipeline in 'automatic mode', with -s3iters 500000, -s4iters 100000, -maxretained 10000, -s2chunksperregion 20, and ploidy set to 1. Convergence was assessed through MCMC traces (*Figure 4—figure supplement 1*) and population assignments (i.e., pairwise coincidence matrix; *Figure 4—figure supplement 1*).

## Population genetic analyses of demography and selection signatures

We computed allele frequencies and estimated the allele spectrum using VCFtools. We used RStudio (*RStudio Team, 2015*) and ggplot2 (*Wickham, 2016*) for visualizations. Synonymous and nonsynonymous sites were identified and annotated using SnpEff v4.3t (*Cingolani et al., 2012*). Variants that were classified by SnpEff as having a 'high', 'moderate', 'low', or 'modifying' impact on the encoded protein sequences. The data was summarized in R using the packages dplyr (*Wickham et al., 2018*), reshape2 (*Wickham, 2007*), tidyr (*Wickham and Henry, 2018*), andggplot2. Furthermore, we performed a genome-wide association study with TASSEL 5 (*Bradbury et al., 2007*) for identifying highly associated SNPs with genetic groups such as S12. Nucleotide diversity was calculated with vcftools -site-pi function using vcf-files that were converted to diploid.

The demographic history of the populations was investigated based on the folded SFS using dadi (*Gutenkunst et al., 2009*). The vcf file was filtered to remove variants predicted to have a 'high' to 'moderate' effect by SnpEff and thinned with a 5 kb distance in order to obtain a set of most likely neutral SNPs. We grouped the isolates of the two smaller clusters of the PCA together (isolates from China, South Korea, Japan, and Georgia) in order to obtain a sufficiently large number of samples. For the European cluster, we kept only one randomly selected S12 isolate to avoid a bias from the large number of S12 clones. We compared the observed SFS against the SFS generated under several demographic models implemented in *dadi* (sub-module dadi.Demographics1D). For all models, we ran the optimization 50 times with initial parameters drawn randomly from the bounded distributions (specified for each model below) and the optimization method *optimiselog* with the maximum number of iterations set to 100 (see *Supplementary file 2* for initial parameters). Models were compared with the AIC. The AIC penalizes models with a higher number of parameters in order to identify the model explaining the largest amount of variation using the fewest possible independent variables. The compared models are described below and include the following parameters: nuF representing the ratio of contemporary to ancient population size, nuB representing the ratio of intermediate population size to ancient population size (relevant for some models only), and T representing the time at which the instantaneous size change happened (in units of 2*Na generations).

- Standard neutral (i.e., constant) model (snm function) without population size change.
- Sudden expansion model (two_epoch function) with nuF bounded between 1 and 1000 and T between 0 and 2000.
- Sudden contraction model (two_epoch function) with nuF bounded between 1 and 1000 and T between 0 and 2000.
- Model implementing a single instantaneous change in population size at a specific point in time followed by exponential growth (bottlegrowth function) with nuF bounded between 0.01 and 1000, nuB between 0.01 and 100, and T between 0 and 500.
- Model implementing a sudden population contraction followed by a sudden expansion (i.e., a bottleneck with recovery; three_epoch function) with nuF bounded between 1 and 2000, nuB between 0.01 and 1, TB (length of the bottleneck) between 0 and 100, and TF (time since the bottleneck) between 0 and 500.

Datasets and scripts for all demographic analyses are accessible here: https://github.com/croll-lab/datasets (*Laboratory of Evolutionary Genetics @ UNINE, 2021*; copy archived at swh:1:rev: fa908efdb71d5ba90823c70bf2467e10ce6f12a7).

Identification of putative selective sweeps was performed using RAiSD v2.5 software that is based on three different genetic signatures of positive selection (with -y 1 -M 0 -w 50 -c 1 parameters) (*Alachiotis and Pavlidis, 2018*). The Manhattan plot shows the distribution of the computed composite $\mu$ statistic of positive selection. To investigate potential false positive selection signals due to the demographic history of the populations, we performed 10,000 ms simulations to model three demographic scenarios in the European population: constant, expansion, and bottleneck. We used the average theta value obtained from the demographic simulations with *dadi* on the two populations for which we could estimate the best fitting demographic history model. The recombination

rate was estimated with LDhat v2.2 (*McVean and Auton, 2007*), with -missfreqcut 0, -samp 2000, -its 5000000, -bpen 20, and -burn 50 (*Figure 3—figure supplement 2*). The average values of phi and of pairwise SNP distance across the 11 scaffolds were used for ms simulations (*Hudson, 2002*). The bottleneck scenario was simulated given the parameters 'ms 10 10000- t 558 -r 0.0032457 12980 -eN .10000000000000000000 0.01'. For the expansion, we considered population doubling 'ms 10 10000 -t 558 -r 0.0032457 12980 -eN .10000000000000000000 2', and the constant population model was performed given identical theta and phi parameters 'ms 10 10000 -t 558 -r 0.0032457 12980'. We performed RAiSD selection scans on each of the three simulated datasets of different demographic scenarios and extracted the highest value of the $\mu$ statistic.

## De novo identification of TEs and effector prediction

We performed a de novo identification of TEs in the *C. parasitica* reference genome EP155 v2.0 using RepeatModeler v2.0.1 (*Flynn et al., 2019*). The consensus sequences were merged with the RepBase library (RepBaseRepeatMaskerEdition-20181026) and then used for repeat annotation using RepeatMasker v4.0.7 with a blast cutoff of 250 (*Smit et al., 2015*). Repeats were then filtered out for low complexity and simple repeats, and parsed using the parseRM_merge_interrupted.pl script from https://github.com/4ureliek/Parsing-RepeatMasker-Outputs. We only retained TEs longer than 100 bp, and overlapping identical TEs were merged into single elements for the final annotation. In addition, TEs of the same family separated by less than 200 bp were considered as part of the same TE and merged into a single element. We analyzed population-level presence/absence variation of TEs using the R-based tool ngs_te_mapper, using trimmed reads as above and bwa version 0.7.17-r1188 (*Bergman, 2012*; *Li and Durbin, 2009*). Overlapping genes and TEs were identified using the intersect function from the bedtools suite v2.29 (*Quinlan and Hall, 2010*). For effector prediction, we first identified putatively secreted proteins for the *C. parasitica* reference genome with SignalP and TMHMM as implemented in InterProScan v5.31–70.0 (*Jones et al., 2014*). Then, protein sequences with a predicted secretion signal predicted by both tools were extracted with samtools v1.9 (*Li et al., 2009*) and used as input for effector prediction with EffectorP v2.0 (*Sperschneider et al., 2018*). EffectorP comes pre-trained on gene sequences encoding functionally confirmed fungal effectors.

## Genome compartmentalization and CNV analyses

We investigated the genome architecture of *C. parasitica* following the protocol described in *Saunders et al., 2014*. Briefly, we computed intergenic distances with the Calculate_FIR_length.pl script using the gene prediction for *C. parasitica* reference genome EP155 v2.0. We defined 40 bins given the range of 3′ and 5′ intergenic distances and calculated the number of genes that fell within each bin. To infer CNV, we computed the normalized read depth (NRD) for all isolates using the CNVcaller pipeline (*Wang et al., 2017*). For this, the reference genome was first split into 800 bp overlapping kmers that were re-aligned to the reference using blasr (-m 5 -noSplitSubreads -minMatch 15 -maxMatch 20 -advanceHalf -advanceExactMatches 10 -fastMaxInterval -fastSDP -aggressiveIntervalCut -bestn 10) to identify duplicated windows (python3 0.2.Kmer_Link.py ref.genome. kmer.aln 800 ref.genome.800.window.link). The resulting windows were then used to calculate the NRD for each isolate from the aligned reads (bam format) in 800 bp windows (Individual.Process.sh -b $bam -h ${i% .bam} -d ref.genome.800.window.link -s scaffold_1). For all further analyses, we used the read depth normalized for the absolute copy number and GC content of each sample (RD_normalized output). Given the observed distribution of the NRD across all isolates, we considered windows with NRD >1.6 to be duplicated and windows with NRD <0.4 as deleted considering the 5th and 95th percentiles of the normalized read coverage (*Figure 5—figure supplement 3*). To investigate the relative diversity within the S12 and non-S12 groups, we resampled 80 random isolates from each lineage 100 times and investigated SNP, TE, and CNV diversity. We quantified the minor allele frequency spectrum in each of the two groups and tested for the group effect (S12 vs. non-S12) on the genetic diversity using Fisher's exact tests in R. We simulated datasets using 2000 resampling replicates to generate expectations for random distributions (*Figure 5—figure supplement 1*).

## Mating experiments

The ability of *C. parasitica* isolates belonging to S12 with mating type MAT-1 to sexually outcross with isolates of opposite mating type was assessed in an inoculation experiment. For this, we randomly selected four S12 and one non-S12 isolate with mating type MAT-1, as well as five isolates of opposite mating type MAT-2 from populations in Italy (Nebrodi), Kosovo (Molliq), and Bosnia (Konic, Vrnograč, Projsa) (*Supplementary file 4*). All isolates of both mating types belonged to the vegetative compatibility type EU-12, and mating pairs were only formed between isolates from the same geographic source populations. As substrate for the pairings, 40-mm-long segments of dormant chestnut (*C. sativa*) stems (15–20 mm in diameter) were split longitudinally in half. The wood pieces were autoclaved twice and placed onto sterile Petri dishes (90 mm diameter), which were filled with 1.5% water agar to enclose the pieces. Mating pairs were inoculated on opposite sides of each halved stem with three replicates per pairing. The inoculated plates were then incubated at 25℃ under a 16 hr photoperiod (2500 lux) for 14 days. After 2 weeks, mating was stimulated by adding 5 mL sterile water to the plates to suspend and distribute the conidia produced by both isolates over the stem segment. Any excess water was subsequently removed and the plates were incubated at 18℃ under an 8 hr photoperiod (2500 lux). After 5 months of incubation, perithecia formation was assessed under a dissecting microscope. To confirm successful outcrossing, single perithecia were carefully extracted from the stromata and crushed in a drop of sterile distilled water. The resulting ascospore suspensions were plated on PDA and incubated at 25℃ for 24–36 hr. Afterwards, single germinating ascospores were transferred to PDA and incubated in the dark for 3–5 days at 25℃. DNA was then extracted from 10 mg of lyophilized mycelium using the kit and instructions by King-Fisher (Thermo Fisher Scientific). All single ascospore cultures were screened for mating types by performing a multiplex PCR following the protocol described in *Cornejo et al., 2019*.

## Acknowledgements

We are grateful to Ursula Oggenfuss for helpful comments on a previous manuscript version. We thank Sandrine Fattore, Hélène Blauenstein, Quirin Kupper, Eva Augustiny, Dario Rüegg, and Silvia Kobel for laboratory assistance. We acknowledge the Genetic Diversity Centre (GDC), ETH Zurich, for technical support and facility access. Ludwig Beenken and Martin Wrann helped with documenting mating experiments. Paolo Cortesi, Michael Milgroom, Kiril Sotirovski, Mihajlo Risteski, Marin Ježić, and Seçil Akilli kindly provided samples. We thank Pierre Gladieux and Nikhil Kumar Singh for discussions and sharing scripts for data analyses. We are grateful for the insightful discussions with Daniel Rigling. Funding was awarded to SP by the Swiss National Science Foundation (grant 170188) and to DC by the Fondation Pierre Mercier pour la science.

## Additional information

### Funding

| Funder | Grant reference number | Author |
|---|---|---|
| Schweizerischer Nationalfonds zur Förderung der Wissenschaftlichen Forschung | 170188 | Simone Prospero |
| Fondation Pierre Mercier pour la science | | Daniel Croll |

The funders had no role in study design, data collection and interpretation, or the decision to submit the work for publication.

### Author contributions

Lea Stauber, Conceptualization, Data curation, Formal analysis, Investigation, Visualization, Writing - original draft, Writing - review and editing; Thomas Badet, Formal analysis, Investigation, Visualization, Writing - review and editing; Alice Feurtey, Conceptualization, Formal analysis, Writing - review and editing; Simone Prospero, Conceptualization, Supervision, Funding acquisition, Writing - review

and editing; Daniel Croll, Conceptualization, Supervision, Funding acquisition, Visualization, Writing - original draft, Writing - review and editing

### Author ORCIDs
Lea Stauber (iD) https://orcid.org/0000-0001-8367-6150
Thomas Badet (iD) https://orcid.org/0000-0001-6130-441X
Simone Prospero (iD) https://orcid.org/0000-0002-9129-8556
Daniel Croll (iD) https://orcid.org/0000-0002-2072-380X

### Decision letter and Author response
Decision letter https://doi.org/10.7554/eLife.56279.sa1
Author response https://doi.org/10.7554/eLife.56279.sa2

## Additional files

### Supplementary files
• Supplementary file 1. Information on all analyzed isolates including sampling location, vegetative compatibility type (EU-type), (SSR) simple sequence repeats genotyping outcome, collector information, collection years, and NCBI accessions.

• Supplementary file 2. Initial and best-fit parameters as well as likelihoods and Akaike information criterion (AIC) values as inferred by *dadi* for demographic analyses. Cluster names correspond to clusters as shown in *Figure 1A*.

• Supplementary file 3. List of the genes in regions with signatures of recent positive selection (based on RAiSD). Putative functions are shown by (PFAM) Protein families (database) and Gene Ontology (GO) annotations. Single nucleotide polymorphisms (SNPs) and the corresponding composite μ statistic are shown.

• Supplementary file 4. Summary of MAT-1 and MAT-2 isolate pairings used for testing sexual recombination in S12 populations. *Isolates from the S12 MAT-1 lineage.

• Transparent reporting form

### Data availability
All raw sequencing data is available on the NCBI Short Read Archive (BioProjects PRJNA604575 and PRJNA644891).

The following datasets were generated:

| Author(s) | Year | Dataset title | Dataset URL | Database and Identifier |
|---|---|---|---|---|
| Stauber L, Prospero S, Croll D | 2020 | Population sequencing of Cryphonectria parasitica | https://www.ncbi.nlm.nih.gov/bioproject/PRJNA604575 | NCBI BioProject, PRJNA604575 |
| Stauber L, Prospero S, Croll D | 2020 | Population sequencing of Cryphonectria species | https://www.ncbi.nlm.nih.gov/bioproject/PRJNA644891 | NCBI BioProject, PRJNA644891 |

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
