## [Decision Letter]

**Acceptance summary:**

This study combines whole genome resequencing of a comprehensive set of natural accessions to provide novel insight into the spread of an important tree disease, chestnut blight. The results provide support for the so-called “bridgehead” effect, a process by which successful invasive lineages arise by recombination among genotypes that are already established in an area of first introduction, and which is thought to be a major driver for alien species invasions.

**Decision letter after peer review:**

Thank you for submitting your article "Emergence and diversification of a highly invasive chestnut pathogen lineage across south-eastern Europe" for consideration by *eLife*. Your article has been reviewed by four peer reviewers, including Vincent Castric as the Reviewing Editor and Reviewer #4, and the evaluation has been overseen by Detlef Weigel as the Senior Editor. The following individual involved in review of your submission has agreed to reveal their identity: Anna-Liisa Laine (Reviewer #2).

The reviewers have discussed the reviews with one another and the Reviewing Editor has drafted this decision to help you prepare a revised submission.

As the editors have judged that your manuscript is of interest, but as described below that additional analyses are required before it is published, we would like to draw your attention to changes in our revision policy that we have made in response to COVID-19 (https://elifesciences.org/articles/57162). First, because many researchers have temporarily lost access to the labs, we will give authors as much time as they need to submit revised manuscripts. We are also offering, if you choose, to post the manuscript to bioRxiv (if it is not already there) along with this decision letter and a formal designation that the manuscript is "in revision at *eLife*". Please let us know if you would like to pursue this option. (If your work is more suitable for medRxiv, you will need to post the preprint yourself, as the mechanisms for us to do so are still in development.)

Summary:

Stauber and colleagues have analysed complete genomes of 188 natural isolates of the chesnut pathogen *Cryphonectria parasitica* across Europe. The results confirm previous accounts that the recent invasion in south-eastern Europe involved a single clonal lineage. The invasive lineage was closely related to nearby European lineages, where outcrossing was predominant. This is in line with the “bridgehead population” hypothesis suggesting that the invasive lineage arose in situ rather than through secondary introduction. In spite of very low nucleotide diversity, the invasive lineage shows some variation in TE content and indels overlapping genic sequences. The manuscript is generally well written and easy to follow.

Essential revisions:

1) Provide an explicit demographic model

– The current study follows a previous analysis (Milgroom et al., 2008) that had largely clarified the invasion scenario in this species. As the manuscript currently stands, the additional insight gained from having sequenced complete genomes is not clearly explained. In particular, beyond the better "marker resolution" as compared to the SCARS markers, polymorphism data gained from complete genomes should enable a comprehensive demographic reconstruction of the invasion scenario. This demographic scenario should then be used as a baseline for all diversity and divergence patterns reported. More generally, a more formal hypotheses testing framework is necessary throughout the paper, as several assertions are made based on the simple observation of patterns, without associated statistical tests.

– This demographic model is especially important for the analysis of natural selection signatures, since these signals are strongly dependent upon specifics of the demographic history.

– Finally, the sweep analysis is based on the non-S12 samples only, so it is unclear how sweeps in the recombinant strains relates to the broader issue of adaptation in the invasive lineage.

2) Modify general framing of the study

– The study presents the scenario under the "bridgehead population effect", but Maynard-Smith et al., 1993, developed a model of epidemic clones, where a successful clone emerges and expands from a recombinant background. While the model was formulated for bacteria, it is similar to the bridgehead effect mentioned here, and should at least be cited and briefly discussed.

– The Abstract claims that the study is aimed to "unravel the mechanisms underpinning the invasion" and to identify the "underlying factors driving invasion", but the results actually provide little insight on these mechanistic aspects, beside clarifying the invasion scenario. At least the underlying hypotheses for these claims should be made more explicit (see below).

– The Introduction covers nicely aspects of adaptation and the importance of genetic diversity. There are indeed examples of invasive events that are triggered by limited genetic diversity, but on the other hand, the analysis by Philibert and colleagues (2011) identified sexual reproduction as one of the three key traits predicting the invasion success of forest pathogens. While it is important to understand the specific event leading to the emergence of an invasive strain, the Introduction could expand a bit on what are the properties of populations that are likely to lead to such events in the first place.

– The Introduction would benefit from some ecological insight. Sexual vs. asexual strategies have ecological consequences relevant for invasive success beyond levels of genetic diversity or mating costs. Sexual vs. asexual life-history stages often vary for example in survival ability during unfavorable environmental periods, in their dispersal abilities, and in environmental cues that trigger germination.

3) Better integrate the genomic analyses

– The structural variation analysis is poorly connected to the main aims of the paper. For instance, the “two-speed genome” hypothesis is interesting, but it is unclear how it fits in the questions addressed in the study. More generally, a precise formulation of hypotheses for the genomic aspects would be needed, as these aspects are presently absent from the Introduction section.

– This part of the manuscript also critically lacks statistical support. For instance, the following assertions “The total TE count variation among the […] non-S12 isolates was larger than the clonal S12 isolates”, “Genes tend to overlap duplications rather than deletions and TEs tend to overall deletions rather than duplications” and “Scaffold 4 and 6 were overall rich in duplications and scaffolds 6 and 7 were rich in deletions” need to be supported by statistical tests.

– Finally, the manuscript repeats at several places that there is a strong contrast between the much lower nucleotide diversity in S12 vs non-S12, but a comparable amount of diversity in TE presence/absence. While this is a potentially interesting observation (raising the question of how TE polymorphisms may behave differently from SNPs), again without formal testing it is currently impossible to evaluate the reality of this contrast.

[Editors' note: further revisions were suggested prior to acceptance, as described below.]

Thank you for submitting your article "Emergence and diversification of a highly invasive chestnut pathogen lineage across south-eastern Europe" for consideration by *eLife*. Your article has been evaluated by Vincent Castric as Reviewing Editor and Detlef Weigel as Senior Editor.

This is a substantial revision, and several samples and new analyses have been added to the study, resulting in clearer patterns. However, the manuscript still suffers from two main shortcomings that need to be addressed, and that are listed below. Please note that we want to see this work published in *eLife*.

– The main concern was the lack of a demographic scenario, possibly confounding the selection analysis. The authors now use the dadi package to calculate the likelihood of three demographic scenarios (stable, bottleneck, expansion) in sets of samples clustered by PCA. However, beside the fact that the Materials and methods and Results are not presented in sufficient details (what exact models were implemented, which parameters do they entail, how were likelihoods compared, what are the values of the ML parameter estimates ?), the demographic reconstruction performs very poorly in the cluster containing the focal European accessions. The reasons for this should at least be discussed (could it be due to the inclusion of a large number of clonal lineages distorting the SFS, in particular S12 ? if so, is there a possibility to remedy this situation ?). This limitation should also be much more clearly acknowledged throughout the manuscript, especially with regard to the selection analysis (Figure 6D).

– The results are still difficult to follow because each analysis is presented in turn, such that the text goes back and forth between the different geographical scales considered. I strongly recommend to 1) start with “global” analyses to identify the four main clusters, 2) then go for the specific analysis of the European cluster, including evidence for recombination, the challenges in attempting to reconstruct its demographic history, and ensuing selection analysis, and finally 3) provide the detailed focus on the invasive S12 lineage, including the co-ancestry analysis, the evidence for recombination within S12, the specific derived variants, TEs and SV they carry, the attempt to reconstruct its possible invasion route and the retention of mating competence. I think this would improve readability by a lot.

[Editors' note: further revisions were suggested prior to acceptance, as described below.]

Thank you for resubmitting your article "Emergence and diversification of a highly invasive chestnut pathogen lineage across south-eastern Europe" for consideration by *eLife*. Your article has been evaluated by Vincent Castric as Reviewing Editor and Detlef Weigel as the Senior Editor.

We would like to draw your attention to changes in our policy on revisions we have made in response to COVID-19 (https://elifesciences.org/articles/57162). Specifically, when editors judge that a submitted work as a whole belongs in *eLife* but that some conclusions require a modest amount of additional new data, as they do with your paper, we are asking that the manuscript be revised to either limit claims to those supported by data in hand, or to explicitly state that the relevant conclusions require additional supporting data.

This revision is a lot clearer, and most of our previous comments have been convincingly dealt with. One important issue remains though with the demographic analysis:

– As pointed in the previous decision letter, obtaining strictly identical maximum likelihood values for all three demographic models implemented is quite unexpected. Unfortunately, the manuscript currently lacks essential details to understand why this happened. For instance, how many iterations of the likelihood optimization process were used (i.e. was the "maxiter" parameter in dadi set to a high enough value to reach convergence ?), how many independent replicates were run ? starting from which (different) initial values ? what parameter bounds were specified ? All these important details are essential but they are lacking from the Materials and methods section. Providing a complete table of all parameters used, along with the dadi input file is essential so that readers can reproduce the analysis.

– I could also not find anywhere in the manuscript how the likelihoods were compared. By a likelihood ratio test ? If so, how were differences in parameter numbers among models taken into account ?

– It is possible that after performing these additional optimization steps the model still fails to converge simply because the unusual shape of the SFS cannot be explained by demography alone, especially in the first cluster. But even in that case it would be essential to report how this conclusion was reached, and to explicitly mention this fact by stating "lack of convergence" in Figure 2 (rather than the strictly identical likelihood values).

Finally, the Discussion contains an assertion that is not supported by the results: "…we found evidence of a demographic expansion consistent with the spread to North America…". In fact, the North-American samples all belong to cluster 1, where the demographic reconstruction is problematic, so this sentence should be rephrased.

---

## [Author Response]

Essential revisions:1) Provide an explicit demographic model– The current study follows a previous analysis (Milgroom et al., 2008) that had largely clarified the invasion scenario in this species. As the manuscript currently stands, the additional insight gained from having sequenced complete genomes is not clearly explained. In particular, beyond the better "marker resolution" as compared to the SCARS markers, polymorphism data gained from complete genomes should enable a comprehensive demographic reconstruction of the invasion scenario. This demographic scenario should then be used as a baseline for all diversity and divergence patterns reported. More generally, a more formal hypotheses testing framework is necessary throughout the paper, as several assertions are made based on the simple observation of patterns, without associated statistical tests.– This demographic model is especially important for the analysis of natural selection signatures, since these signals are strongly dependent upon specifics of the demographic history.

We have now revised our manuscript in multiple ways including new datasets and new analyses to address the issues highlighted above.

Additional samples: Our manuscript documents how *C. parasitica* first colonized Europe to form a bridgehead for a secondary invasion by the S12 lineage across southeastern Europe. To more accurately assess the impact on the pathogen diversity after the arrival in Europe, we added samples across the global distribution range including from China, South Korea, Japan and the USA (n = 25 add for a total set of genomes in the study of 230). We believe that the US source population is still under-sampled, however the additional samples provide now a more robust framework to estimate the demography of this species and the history of the European colonization. The new samples are now integrated in the first part of the analyses, before focusing specifically on the S12 lineage. Figures and tables were updated accordingly.

Demographic modelling: We performed demographic modelling as requested to provide a more formal framework to investigate the history of the European invasion. Analyzing the global set of isolates (excluding all but one S12 clonal genotype). Our analyses on the global collection showed signatures of recent population expansions consistent with the spread to the United States and more recently Europe. We could not find a good-enough fit in the narrower group of European/North American isolates, as the observed folded site frequency spectrum (SFS) of the European/North American group is W-shaped, instead of a steadily decline curve (as found in the simulated data). This unusual frequency spectrum likely stems from many quasi-clones from several groups and inbreeding. We show now all demographic model fits in Figure 2.

The selection analyses in the non-S12 European population has now also been revised to use demography as a baseline thus increasing the confidence that the genomic signatures observed in this analysis are indeed due to selection. Notably, we used simulated populations under constant size, bottleneck, expansion scenarios, to determine a threshold above which signals can confidently be assigned to selection in the *C. parasitica* data set.

– Finally, the sweep analysis is based on the non-S12 samples only, so it is unclear how sweeps in the recombinant strains relates to the broader issue of adaptation in the invasive lineage.

We have indeed performed the selection analyses on the European population. Our rationale was to identify recent signatures of selection important for the colonization of the European habitats. Performing the selection scans on the clonal set of genotypes of the S12 would not be possible due the very high degree of linkage disequilibrium and extremely low diversity. We have previously performed an analysis on the frequency spectrum of the accumulated mutations and their putative impact. We found low-frequency deleterious mutations segregating in the S12 lineage.

2) Modify general framing of the study– The study presents the scenario under the "bridgehead population effect", but Maynard-Smith et al., 1993, developed a model of epidemic clones, where a successful clone emerges and expands from a recombinant background. While the model was formulated for bacteria, it is similar to the bridgehead effect mentioned here, and should at least be cited and briefly discussed.

We are grateful for this request and have now modified the Introduction when discussing the conceptually related model of the "bridgehead effect".

– The Abstract claims that the study is aimed to "unravel the mechanisms underpinning the invasion" and to identify the "underlying factors driving invasion", but the results actually provide little insight on these mechanistic aspects, beside clarifying the invasion scenario. At least the underlying hypotheses for these claims should be made more explicit (see below).

We have now clarified our aims to primarily unravel the invasion history of chestnut blight in south-eastern Europe. Moreover, we have better integrated the genomic results that highlight the genetic architecture of chestnut blight.

– The Introduction covers nicely aspects of adaptation and the importance of genetic diversity. There are indeed examples of invasive events that are triggered by limited genetic diversity, but on the other hand, the analysis by Philibert and colleagues (2011) identified sexual reproduction as one of the three key traits predicting the invasion success of forest pathogens. While it is important to understand the specific event leading to the emergence of an invasive strain, the Introduction could expand a bit on what are the properties of populations that are likely to lead to such events in the first place.

We now broadly introduce critical factors linked to successful invasions include mode of reproduction (both the advantage of sex but also the advantage of switching to clonal reproduction depending on the circumstances and environment).

– The Introduction would benefit from some ecological insight. Sexual vs. asexual strategies have ecological consequences relevant for invasive success beyond levels of genetic diversity or mating costs. Sexual vs. asexual life-history stages often vary for example in survival ability during unfavorable environmental periods, in their dispersal abilities, and in environmental cues that trigger germination.

We have now added a more ecological perspective to the Introduction in particular related to environmental constraints, spore production and dispersal.

3) Better integrate the genomic analyses– The structural variation analysis is poorly connected to the main aims of the paper. For instance, the “two-speed genome” hypothesis is interesting, but it is unclear how it fits in the questions addressed in the study. More generally, a precise formulation of hypotheses for the genomic aspects would be needed, as these aspects are presently absent from the Introduction section.

We have now expanded the Introduction to cover genomic insights into pathogen emergence and set up hypotheses to be tested.

– This part of the manuscript also critically lacks statistical support. For instance, the following assertions “The total TE count variation among the […] non-S12 isolates was larger than the clonal S12 isolates”, “Genes tend to overlap duplications rather than deletions and TEs tend to overall deletions rather than duplications” and “Scaffold 4 and 6 were overall rich in duplications and scaffolds 6 and 7 were rich in deletions” need to be supported by statistical tests.

We have now added a series of statistical tests to the genomic analyses to provide a clearer statistical framework and enable hypothesis testing.

– Finally, the manuscript repeats at several places that there is a strong contrast between the much lower nucleotide diversity in S12 vs non-S12, but a comparable amount of diversity in TE presence/absence. While this is a potentially interesting observation (raising the question of how TE polymorphisms may behave differently from SNPs), again without formal testing it is currently impossible to evaluate the reality of this contrast.

We address now these differences in a quantitative way and provide statistical support where this is feasible. We now also briefly discuss these observations.

[Editors' note: further revisions were suggested prior to acceptance, as described below.]

This is a substantial revision, and several samples and new analyses have been added to the study, resulting in clearer patterns. However, the manuscript still suffers from two main shortcomings that need to be addressed, and that are listed below. Please note that we want to see this work published in eLife.– The main concern was the lack of a demographic scenario, possibly confounding the selection analysis. The authors now use the dadi package to calculate the likelihood of three demographic scenarios (stable, bottleneck, expansion) in sets of samples clustered by PCA. However, beside the fact that the Materials and methods and Results are not presented in sufficient details (what exact models were implemented, which parameters do they entail, how were likelihoods compared, what are the values of the ML parameter estimates ?), the demographic reconstruction performs very poorly in the cluster containing the focal European accessions. The reasons for this should at least be discussed (could it be due to the inclusion of a large number of clonal lineages distorting the SFS, in particular S12 ? if so, is there a possibility to remedy this situation ?).

We added the requested details about the modelling to the Materials and methods section. We also added a supplementary table (Supplementary file 2) including more details about the results from the analyses including the best-fit parameters and likelihoods.

We now more explicitly state in the Materials and methods and Results that we have excluded all S12 isolates (except one) for demographic analyses. This has also always been the case for the selection scans. We regret that our wording was not clear enough in this regard. Hence, the large number of S12 clone genotypes in our overall dataset never had an impact on either demographic or selection analyses.

Additionally, we now briefly discuss possible reasons for the poor demographic reconstruction within the focal European cluster. Even after excluding S12 genotypes, the pairwise genetic distances between genotypes are somewhat bimodal with some fairly close genotypes of possible clonal origin.

This limitation should also be much more clearly acknowledged throughout the manuscript, especially with regard to the selection analysis (Figure 6D).

We now state more explicitly the limitation of the demographic reconstruction with regard to the selection analysis.

– The results are still difficult to follow because each analysis is presented in turn, such that the text goes back and forth between the different geographical scales considered. I strongly recommend to 1) start with “global” analyses to identify the four main clusters, 2) then go for the specific analysis of the European cluster, including evidence for recombination, the challenges in attempting to reconstruct its demographic history, and ensuing selection analysis, and finally 3) provide the detailed focus on the invasive S12 lineage, including the co-ancestry analysis, the evidence for recombination within S12, the specific derived variants, TEs and SV they carry, the attempt to reconstruct its possible invasion route and the retention of mating competence. I think this would improve readability by a lot.

We are grateful for this suggestion. We have now changed the order of the Results to:

– Global analyses: Identification of clusters

– Demographic analyses: for global and European cluster (including challenges for demographic reconstruction)

– European analyses: phylogenetic network, recombination and selection analyses

– S12 analyses: Coancestry, TE and SV, variants in S12 and retention of mating competence

[Editors' note: further revisions were suggested prior to acceptance, as described below.]

This revision is a lot clearer, and most of our previous comments have been convincingly dealt with. One important issue remains though with the demographic analysis:– As pointed in the previous decision letter, obtaining strictly identical maximum likelihood values for all three demographic models implemented is quite unexpected. Unfortunately, the manuscript currently lacks essential details to understand why this happened. For instance, how many iterations of the likelihood optimization process were used (i.e. was the "maxiter" parameter in dadi set to a high enough value to reach convergence ?), how many independent replicates were run ? starting from which (different) initial values ? what parameter bounds were specified ? All these important details are essential but they are lacking from the Materials and methods section. Providing a complete table of all parameters used, along with the dadi input file is essential so that readers can reproduce the analysis.

We are now stating more clearly in the Materials and methods section what parameters were used for demographic inference, including the number of iterations, the number of independent replicates and parameter bounds. We have also updated the Supplementary file 2, which now also includes the initial parameters and AIC values for model comparisons. Moreover, data sets and scripts for demographic analyses (including the genetic dataset) can now also be found at: https://github.com/crolllab/datasets (link also added to the Materials and methods section).

– I could also not find anywhere in the manuscript how the likelihoods were compared. By a likelihood ratio test ? If so, how were differences in parameter numbers among models taken into account ?

We are thankful for this comment. We have now implemented the Akaike Information Criterion (AIC) for model comparisons. This is now clearly stated in the Materials and methods sections. Moreover, we provide a full list of AIC values for all models in Supplementary file 2.

– It is possible that after performing these additional optimization steps the model still fails to converge simply because the unusual shape of the SFS cannot be explained by demography alone, especially in the first cluster. But even in that case it would be essential to report how this conclusion was reached, and to explicitly mention this fact by stating "lack of convergence" in Figure 2 (rather than the strictly identical likelihood values).

We now state more explicitly that no parameter optimization was reached for the CL1 cluster.

Finally, the Discussion contains an assertion that is not supported by the results: "…we found evidence of a demographic expansion consistent with the spread to North America…". In fact, the North-American samples all belong to cluster 1, where the demographic reconstruction is problematic, so this sentence should be rephrased.

We fully agree and have decided to delete the sentence entirely.